# Particles' phase state variability in the North Atlantic free troposphere during summertime determined by different atmospheric transport patterns and sources

Zezhen Cheng[1], Megan Morgenstern[2], Bo Zhang[3], Matthew Fraund[4], Nurun Nahar Lata[1], Rhenton Brimberry[1], Matthew A. Marcus[4], Lynn Mazzoleni[2], Paulo Fialho[5], Silvia Henning[6], Birgit Wehner[6], Claudio Mazzoleni[2], Swarup China[1]

[1] Environmental Molecular Sciences Laboratory, Pacific Northwest National Laboratory (PNNL), Richland, Washington 99352, USA

[2] Atmospheric Sciences Program, Michigan Technological University, Houghton, Michigan, 49921, USA

[3] National Institute of Aerospace, Hampton, VA 23666, USA

[4] Advanced Light Source, Lawrence Berkeley National Laboratory, Berkeley, CA 94720, USA

[5] Institute of Volcanology and Risk Assessment – IVAR, Rua da Mãe de Deus, 9500-321 Ponta Delgada, Portugal

[6] Leibniz Institute for Tropospheric Research, Permoserstraße 15, 04318 Leipzig, Germany

*Correspondence to*: Swarup China (Swarup.china@pnnl.gov)

**Abstract.** Free tropospheric aerosol particles have important but poorly constrained climate effects due to transformations of their physicochemical properties during long-range transport. In this study, we investigated the chemical composition and provided an overview of the phase state of individual particles that have been long-range transported over the North Atlantic Ocean in June and July 2014, 2015, and 2017 to the Observatory of Mount Pico (OMP), in the Azores. OMP is an ideal site for studying long-range transported free tropospheric particles with negligible influence from local emissions and rare contributions from the boundary layer. We used the FLEXible PARTicle Lagrangian particle dispersion model (FLEXPART) to determine the origin and transport trajectories of sampled air masses and found that most originated from North America and recirculated over the North Atlantic Ocean. The FLEXPART analysis shows that the sampled air masses were highly aged (average plume age >10 days). Size-resolved chemical compositions of individual particles were probed using computer-controlled scanning electron microscopy with an energy dispersive X-ray spectrometer (CCSEM-EDX) and scanning transmission X-ray microscopy with near-edge X-ray absorption fine structure spectroscopy (STXM-NEXAFS). CCSEM-EDX results show that the most abundant particle types were carbonaceous (~29.9 to 82.0 %), sea salt (~0.3 to 31.6 %), and sea salt with sulfate (~2.4 to 31.5 %). We used a tilted stage interfaced within an Environmental Scanning Electron Microscope (ESEM) to determine the phase state of individual submicron particles. We found that most particles (~47 to 99 %) were in the liquid state at the time of collection due to inorganic inclusions. Moreover, we also observed a substantial fraction of solid

and semisolid particles (~0 to 30 % and ~1 to 42 %, respectively) during different transport patterns/events, reflecting the particles' phase state variability for different atmospheric transport events and sources. Combining phase state measurements with FLEXPART CO tracer analysis, we found that wildfire-influenced plumes can result in particles with a wide range of viscosities after long-range transport in the free troposphere. We also used temperature and RH values extracted from the Global Forecast System (GFS) along the FLEXPART simulated path to predict the phase state of the particles during transport

and found that neglecting internal mixing with inorganics would overestimate the viscosity of free tropospheric particles. Our findings warrant future investigation on the quantitative assessment of the influence of internal mixing on the phase state of the individual particles. This study also provides insights into the chemical composition and phase state of free tropospheric particles, which can benefit models to reduce uncertainties in ambient aerosol particles' effects on climate.

**Short summary.** We observed a high abundance of liquid and internally mixed particles in samples collected in the North Atlantic free troposphere during summer. We also found several solid and semisolid particles for different emission sources and transport patterns. Our results suggest that considering the mixing state, emission source, and transport patterns of particles is necessary to estimate their phase state in the free troposphere, which is critical for predicting their effects on climate.

## 1 Introduction

Atmospheric aerosol particles play a vital role in regional and global climates. They influence climate by interacting with solar and terrestrial radiation and by interacting with clouds when acting as cloud droplet condensation and heterogeneous ice nuclei (Bellouin et al., 2020; Bond et al., 2013; Fan et al., 2016; Laskin et al., 2015; Moosmüller et al., 2009; Saleh, 2020). The efficiency of atmospheric particles acting as cloud droplets or ice nuclei strongly depends on their chemical composition and physical properties (Fan et al., 2016; IPCC, 2013; Seinfeld et al., 2016). However, the current understanding of atmospheric

aerosols' climate effects is still limited. One reason for such a gap in our understanding is that atmospheric aerosols have a wide range of viscosities and exhibit different phase states (e.g., solid (viscosity $>10^{12}$ Pa s), semisolid ($10^2$ to $10^{12}$ Pa s), and liquid ($<10^2$ Pa s)) (Reid et al., 2018; Virtanen et al., 2010, 2011). The viscosity affects the ability of the particle to participate in several important atmospheric processes such as cloud condensation (Hodas et al., 2015; Liu et al., 2018b), heterogeneous ice nucleation (more viscous/solid particle can promote ice formation) (Berkemeier et al., 2014; Knopf et al., 2018; Murray et

al., 2010; Reid et al., 2018; Sharma et al., 2018), and atmospheric aging process (e.g., higher viscosity reduces atmospheric reactivity and therefore slows aging) (Berkemeier et al., 2016; Kuwata and Martin, 2012; Liu et al., 2018a; Marshall et al., 2016; Pöschl and Shiraiwa, 2015a; Renbaum-Wolff et al., 2013). Several studies have found that the viscosity of atmospheric aerosols is affected by factors such as emission source, formation mechanism, material properties, chemical compositions, and ambient conditions (Koop et al., 2011; Li et al., 2020; Shiraiwa et al., 2017; Shrivastava et al., 2017; Virtanen et al., 2010).

Several studies have focused on the viscosity and/or phase state of atmospherically relevant submicron-sized particles (Bateman et al., 2014, 2015; Hosny et al., 2016; Jain and Petrucci, 2015; Li et al., 2017; Pajunoja et al., 2016; Renbaum-Wolff

et al., 2013; Virtanen et al., 2010), but only a few studies reported field measurements of the phase state of ambient particles (Bateman et al., 2016, 2017; Liu et al., 2017, 2019; Pajunoja et al., 2016; Slade et al., 2019), and even more sparse are studies of aerosol phase state at high-altitude in the Free Troposphere (FT). This data gap is due to limitations in the measurement techniques. Such limitations stem from the low particle concentrations at remote sites, particle sizes beyond the instrument's detection range, and challenges in directly measuring the phase state of individual particles. These drawbacks can be addressed by collecting aerosols on substrates and applying offline aerosol phase state analysis. One example of such analysis is using tilted SEM imaging to determine the phase state of particles based on the shape they acquire upon impaction on the substrate. In fact, solid, semisolid, and liquid organic particles will deform to near spheric, dome-like, and flat shapes, respectively, when they impact on the substrates (Cheng et al., 2021; Reid et al., 2018; Sharma et al., 2018; Wang et al., 2016). Based on this approach, we applied a new analytical platform that uses tilted Environmental Scanning Electron Microscope (ESEM) imaging to directly observe and assess the phase state of particles based on their shape deformation on the substrate (Cheng et al., 2021).

The FT is the atmospheric layer that extends above the planetary boundary layer (PBL), from about 1 km altitude to the tropopause (Seinfeld and Pandis, 2006). Particles are injected into the FT mainly through convection and frontal uplift or are formed by oxidation of precursor gases (North et al., 2014). In turn, FT aerosols can be transported to the PBL via entrainment (De Wekker and Kossmann, 2015), dry deposition (Zufall and Davidson, 1998), and dry intrusions, which are events in which cold, dry air rapidly descends from the FT down to the PBL (Ilotoviz et al., 2021; Raveh-Rubin, 2017; Raveh-Rubin and Catto, 2019; Tomlin et al., 2021). The chemical composition of FT aerosols is complex (Bondy et al., 2018; Cozic et al., 2008; Dzepina et al., 2015; Schum et al., 2018; Zhou et al., 2019) and continuously evolves since FT aerosols typically have longer lifetimes and are transported over long distances before reaching remote locations (Cozic et al., 2008; Gogoi et al., 2014; Haywood and Boucher, 2000; Huang et al., 2008). Thus, FT aerosols have longer atmospheric aging times to experience more physical interactions and chemical reactions with other atmospheric components, leading to more complex physical and chemical properties (Dunlea et al., 2009; Gogoi et al., 2014; Huang et al., 2008; Jaffe et al., 2005; Laing et al., 2016; Sun et al., 2009; Zhou et al., 2019). Several previous studies focused on FT aerosols at remote sites, improving our understanding of regional and long-range transported aerosols (Boose et al., 2016; China et al., 2015, 2017; Clarke et al., 2013; Dzepina et al., 2015; Rinaldi et al., 2015; Rose et al., 2017; Schum et al., 2018; Zhou et al., 2019). For instance, Dzepina et al., 2015 conducted a study at the Observatory of Mount Pico (OMP) during the summer of 2012, and they found that FT aerosols underwent atmospheric oxidation such as photooxidation and aqueous-phase reactions (e.g., cloud processing) during the long-range transport, and evaluated their chemical and physical properties. Moreover, Zhou et al., 2019 reported that organic aerosols in the FT were more oxidized and less volatile than those in the boundary layer, based on a study at the Mount Bachelor Observatory. Similarly, after analyzing samples collected at OMP during summertime, Schum et al., 2018 found that FT organic aerosols might be more viscous than PBL aerosols due to lower ambient temperature and relative humidity (RH) values, suggesting that FT organic aerosols might be less reactive than PBL organic particles. FT aerosols also indirectly affect

climate by changing clouds' albedo since they can act as cloud condensation nuclei (CCN) and ice nucleating particles (INP) (China et al., 2017; Clarke et al., 2013; Rose et al., 2017), and their chemical composition, particle size, mixing state, and phase state can affect their roles as CCN and INP (Ching et al., 2017; King et al., 2012; Knopf et al., 2018; Murray et al., 2010; Riemer et al., 2019; Schmale et al., 2017; Wang et al., 2010). Therefore, a better understanding of FT aerosols' chemical composition and phase state would improve the estimation of aerosol climate effects in current models. In this study, we focus on the chemical composition and phase state of FT aerosol and their correlation.

High-altitude mountaintop observatories are often used to study FT particles and atmospheric chemistry with advantages with respect to airborne field studies such as lower costs and long-term continuous measurements (Zhou et al., 2019). Various mountaintop sites are operating all over the world to investigate the properties of FT aerosols and trace gases, such as the High Altitude Research Station Jungfraujoch, Switzerland (Bianchi et al., 2016; Boose et al., 2016; Cozic et al., 2008; Motos et al., 2020), Mt. Cimone station in Italy (Fischer et al., 2003; Marinoni et al., 2008; Rinaldi et al., 2015), Mount Bachelor Observatory in central Oregon, US (Briggs et al., 2016; Weiss-Penzias et al., 2006; Zhou et al., 2019), and the OMP in the North Atlantic Ocean (Dzepina et al., 2015; Val Martin et al., 2008b). Along with these sites, the OMP is an ideal site for studying long-range transported FT aerosols from North America to Europe with negligible contribution from local sources (Val Martin et al., 2008b, 2008a) and is one of the sites that is least influenced by the PBL (Coen et al., 2018). OMP is located in the summit caldera of the Pico Volcano (2225 m above sea level) on Pico Island in the Azores archipelago in the North Atlantic. The influence of local emission and marine boundary layer during summer on the air masses sampled at the site are typically negligible, making OMP an excellent location for long-range transported FT particles above the North Atlantic Ocean (Schum et al., 2018).

In this study, we present an overview of the overall phase state of individual FT atmospheric aerosol particles (internally mixed) collected at OMP over three different years, which are July 2014 (Pico 2014), June and July 2015 (Pico 2015), and 2017 (Pico 2017). Analysis of samples from three years using tilted view ESEM imaging and scanning transmission X-ray microscopy with near-edge X-ray absorption fine structure spectroscopy (STXM-NEXAFS) are reported to study the phase state of individual particles. The chemical composition and phase state of individual particles for Pico 2014 have been reported in a previous study (Lata et al., 2021). The chemical composition of individual particles for Pico 2015 will be discussed in future work. This study focuses on detailed individual particle analysis of the Pico 2017 samples. We performed FLEXible PARTicle Lagrangian particle dispersion model (FLEXPART) simulations to select specific events to retrieve the origin of sampled air masses, transport trajectories, atmospheric aging time, and temperature and relative humidity in the air masses along the transport path. We applied multi-modal micro-spectroscopy techniques to probe aged and long-range transported particles' chemical composition and phase state. This study highlights how FT particles' phase state varies during different atmospheric transport events by combining probabilistic FLEXPART back trajectory analysis.

## 2 Experiment

### 2.1 Sampling site and sample collection

Samples of atmospheric particles were collected at the OMP, which is located in the summit caldera of the Pico Volcano (at 2225 m above sea level, 38.47° N, 28.40° W) in the Azores, Portugal (Dzepina et al., 2015; Honrath et al., 2004; Val Martin et al., 2008b). Pico 2014 and Pico 2015 samples were collected during July 2014 and June and July 2015 at OMP, and details of sampling times and conditions are listed in Table S1. Pico 2017 samples were collected during June and July 2017, and Table S2 reports the sampling times and environmental conditions. Available hourly temperature and RH variation are shown in Fig. S2. Additional experimental details are provided in the supporting information. Sample collection and storage for three years Pico samples followed the same protocol. All samples were collected on TEM B-film grids (300 mesh, Ted Pella, Inc) and lacey formvar grids (300 mesh, Ted Pella, Inc.) using a four-stage cascade impactor (MPS-4G1) at a flow rate of ~7 lpm for multi-modal micro-spectroscopy analysis. Particles were collected on the third and/or fourth stages of the impactor with 50 % cutoff aerodynamic diameters of >0.15 μm and >0.05 μm, respectively (see Table S1 and S2). Samples were placed in dedicated storage boxes wrapped in Al foil and kept in zip lock bags immediately after collection to avoid exposure to light and outside air, which is a typical sample storage strategy for field-collected aerosol samples (e.g., Adachi and Buseck, 2011; Kirillova et al., 2016; Marsh et al., 2017; Stockwell et al., 2016). The samples were then stored at ambient conditions to reduce the chances of particle modifications and oxidation that might have partially intercurred. However, we cannot exclude with certainty that some of such transformations might have taken place between the sampling and the analysis times. This is a limitation of any offline analysis of field-collected samples. We underline that the site is quite difficult to access; therefore, samples were delivered and analyzed as soon as it was feasible (less than one year after collection). Moreover, from 05 July to 21 July 2017, we also deployed a Scanning Mobility Particle Sizer (SMPS, TROPOS, for details, see Wiedensohler et al., 2012 ) coupled with a silica gel diffusion dryer to monitor the dry particle size distribution (<40% RH) and the total particle concentration with 5 mins time resolution (Siebert et al., 2021).

### 2.2 Origin of sampled air masses using FLEXPART

The FLEXPART was used to determine the origin of sampled air masses and their transport trajectories to OMP (Owen and Honrath, 2009; Seibert and Frank, 2004; Stohl et al., 2005). FLEXPART backward simulations were driven by meteorology fields of the Global Forecast System (GFS) featured with 3-hourly temporal resolution, 1° horizontal resolution, and 26 vertical levels. The output was saved in a grid with a horizontal resolution of 1° latitude by 1° longitude and eleven vertical levels from the surface to 15,000 m a.s.l. More details of the model configurations can be found in (Zhang et al., 2014, 2017). FLEXPART simulated the spatial distribution of upwind residence time of the observed air masses. Transport trajectories (also called a "retroplume") of air masses were obtained by integrating the matrices of residence time over time and altitudes. We calculated a FLEXPART CO tracer by multiplying FLEXPART residence time with CO emission inventories from the Emissions Database for Global Atmospheric Research (EDGAR, version 3.2 (Olivier and Berdowski, 2001)) and the Global Fire

Assimilation System (Kaiser et al., 2012) to estimate influence from anthropogenic and wildfire sources, respectively (Dzepina et al., 2015). We also extracted ambient temperature and relative humidity along the FLEXPART simulated transport pathways by sampling meteorological conditions in the GFS fields for each aerosol sample. The ambient temperature and relative humidity were used to estimate the aerosol phase state during transport (Schum et al., 2018).

**2.3 Micro-spectroscopy analysis of individual particles**

The particles collected from OMP were analyzed with a STXM-NEXAFS and a computer-controlled scanning electron microscopy with energy dispersive X-ray spectroscopy (CCSEM-EDX) to probe their physicochemical properties (Laskin et al., 2005, 2006). We utilized an ESEM (Quanta 3D, Thermo Fisher) equipped with a FEI Quanta digital field emission gun, operated at 20 kV and 480 pA. Ambient particle samples were analyzed with ESEM at 293 K, under vacuum conditions (~2×10-6 Torr) and therefore at RH values near zero, which might lead to losses of volatile and semivolatile materials. Moreover, the temperature and RH inside the ESEM chamber differed from those at the OMP during sample collections (about 10 K higher and 6-67% lower, respectively, see Fig. S2). RH and T affect the phase state of airborne particles; however, our inference of the particle's phase state at the time of collection is based on the shape the particle acquires at impaction on the substrate, which unlikely would change significantly within the ESEM chamber due to adhesion forces between the particle and the substrate. These limitations need to be considered when interpreting our results. ESEM images were used to retrieve individual particles' shape, morphology, and projected size (area equivalent diameter). The CCSEM, equipped with an EDX spectrometer (EDAX, Inc.), and the EDX spectra of individual particles were used to determine the chemical composition of the particles by quantifying the relative element percentage of 15 elements (C, N, O, Na, Mg, Al, Si, P, S, Cl, K, Ca, Mn, Fe, Zn). Since the EDX analysis of light elements such as C, N, and O is considered semi-quantitative and there are C and O contributions from the B-film substrate, we performed post-correction on the element percentage of C, N, and O (see SI Sect. S1 and Fig. S1). The corrected average element percentage of 15 elements of all samples is shown in Fig. S3. As shown in Fig. S3, all samples are dominated by C, N, and O (average element percentage of C, N, and O for each sample are 51.3 to 88.6 %, 0.6 to 22.0 %, 7.5 to 27.9 %, respectively), and the microscopy images show a minor fraction of soot particles, suggesting most particles are organic-rich. Based on their element percentage, each particle in Pico 2017 can be classified as organic (OC), carbonaceous with nitrogen (CNO), carbonaceous with sulfate (CNOS), sea salt (Na-rich), sea salt with sulfate (Na-rich with S), dust (Al, Si, Ca, Fe), dust with sulfate (Al, Si, Ca, Fe, S), and others (see Fig. S4). CCSEM-EDX based particle classification for Pico 2014 can be found in Lata et al., 2021, and that for Pico 2015 will be discussed in our future work.

We used the STXM-NEXAFS spectroscopy at beamline 5.3.2.2 of the Advanced Light Source (ALS) at the Lawrence Berkley National Laboratory to probe the chemical bonding of carbon functional groups of individual particles. Due to beamline time constraints for STXM analysis, we focused only on selected samples and a limited number of selected particles (653 for SA1, 208 for SA2, and 425 for SA3 for Pico 2014, 86 for S3 and 37 for S5 for Pico 2015, and 140 and 166 particles for S3-3 and

S4-2 for Pico 2017). STXM uses a focused monochromatic soft X-ray beam generated from the synchrotron light source using a zone plate with 25 nm outer zones. The sample is rastered under the beam, and the transmitted intensity is recorded to create an image. Spectroscopy is done by repeating this process at 111 energies to create a 'stack', in which each pixel contains a transmission spectrum (Moffet et al., 2011). STXM-NEXAFS data was also used to calculate the total carbon absorbance (TCA) and the organic volume fraction (OVF). TCA at each pixel was calculated according to Eq. (1):

$$TCA = OD_{320} - OD_{278}, \tag{1}$$

where $OD_{320}$ and $OD_{278}$ are the optical density (OD) of post- (320 eV) and the pre- (278 eV) carbon K-edge, respectively, and the TCA of individual particles was calculated as an average over each particle (O'Brien et al., 2014). Since OD was calculated as:

$$OD = -\ln\left(I/I_0\right) = \mu\rho t, \tag{2}$$

Where $I_0$ is the incident X-ray radiation intensity, $I$ is the transmitted intensity, $\mu$ is the mass absorption coefficient (cm$^2$ g$^{-1}$), $\rho$ is the density (g cm$^{-3}$), and $t$ is the thickness of the particle (cm). Thus, TCA is proportional to the particle thickness, and it can be used as an indicator for particle thickness (O'Brien et al., 2014; Tomlin et al., 2020). The OVF was calculated based on a previously published method (Fraund et al., 2019). Briefly, we assume that each pixel in a particle is a mixture of elemental carbon (EC), organic (OC), and inorganic components (IN). Thus, the OVF at each pixel can be calculated as the ratio of the thickness of the organic ($t_{OC}$) divided by the total thickness of that pixel (sum of the thickness of the OC, IN, and EC: $t_{OC}+t_{IN}+t_{EC}$), and the OVF of each particle is the average over the different pixels within each particle (Fraund et al., 2019; Moffet et al., 2010; Pham et al., 2017). Based on Eq. 2, OD at pre- and post- edge and the sp$^2$ peak (285.4 eV) were used to calculate the thickness of OC, IN, and EC, respectively (Fraund et al., 2019). Due to the variability of the $\mu$ and $\rho$ values for the different OC and inorganic species, we used (NH$_4$)$_2$SO$_4$ as a surrogate for the IN of the particles based on the particles' average elemental composition retrieved from CCSEM-EDX measurements (see Sect. 3.2.1), and oxalic acid (C$_2$H$_2$O$_4$) as a surrogate for the organic components since oxalic acid has been shown to be abundant in ambient organic particles and has a representative oxygen-to-carbon ratio similar enough to ambient organic particles' (Fraund et al., 2019; Sorooshian et al., 2006; Yamasoe et al., 2000).

In addition, OD values at four critical energies (Pre-edge, sp$^2$, -COOH (288.5 eV), and post-edge) acquired from STXM spectra were used to classify regions in an individual particle as OC ($OD_{288.5} - OD_{278} > 0$), IN ($OD_{278} / OD_{320} > 0.5$), and EC ($OD_{285.4} /(OD_{320} - OD_{278}) > 0.35$) based on a previously published method (Fraund et al., 2019; Moffet et al., 2010, 2013). We classified each particle into 6 typical classes based on the volume fraction of OC, EC, and IN components: 1) OC-rich (OC is greater than 96 % and both EC and IN are less than 2 %); 2) EC-rich (EC is greater than 96 % and both OC and IN are less than 2 %); 3) IN-rich (IN is greater than 96 % and both OC and EC are less than 2 %); 4) OCEC (both OC and EC are greater than 2 %, and IN is less than 2 %); 5) OCIN (both OC and IN are greater than 2 %, and EC is less than 2 %); and 6) OCINEC (OC, EC, and IN are all greater than 2 %).

**2.4 Tilted imaging**

We utilized tilted view imaging combined with the ESEM to estimate the phase state of particles based on their shapes. For each sample, we evaluated more than 150 randomly selected particles. Moreover, tilted view imaging and CCSEM-EDX experiments were performed independently. Typically, the shapes of solid, semisolid, and liquid organic particles are near spherical, dome-like, and flat, respectively, when they impact on the surface of a substrate (Reid et al., 2018). For organic materials with solid inorganics (e.g., dust, soot, and salt) inclusion, the particle shape might be irregular and not follow the corrected aspect ratio threshold we proposed for organics (Cheng et al., 2021). Therefore, this study investigates only the phase state of organic particles that are not irregular in shape. To quantitatively assess the phase state of particles, we use the particles' corrected aspect ratio (corrected particle width/height ratio ($W_{corrected}/H_{corrected}$)) retrieved from tilted SEM images. A higher aspect ratio means that the particle deformed significantly upon impaction and is more liquid (less viscous) (Cheng et al., 2021; Reid et al., 2018; Sharma et al., 2018; Wang et al., 2016). In this study, we used the same method introduced in Cheng et al., 2021 to retrieve the corrected aspect ratio of the particles. Briefly, we first calculated the tilted aspect ratio using the maximum particle length in the horizontal and vertical direction in the tilted ESEM images ($W_{tilted}$ and $H_{tilted}$, respectively) at a tilted view angle of 75° (tilted aspect ratio $=(W_{tilted}/H_{tilted})$); where $H_{tilted}$ is the projection of the arc from top to base on the horizontal plane, and $W_{tilted}$ is the projection of base width on the horizontal plane, which is equal to the $W_{corrected}$. The shape will appear circular in the tilted SEM images for solid spherical organic particles ($W_{tilted}$ and $H_{tilted}$ are equal to $W_{corrected}$ and $H_{corrected}$, respectively). Thus, the tilted aspect ratio is equal to the corrected aspect ratio for solid spherical particles. However, the shapes of semisolid and liquid organic particles will be distorted in the vertical direction ($W_{tilted}=W_{corrected}$, but $H_{tilted}>H_{corrected}$). When the particle is extremely flat (liquid state and very low viscosity), the $H_{tilted}$ is the projection of the vertical diameter of the base on the horizontal plane. Thus, in this case, the measured tilted aspect ratio is greater than or equal to 1/cos(tilt angle) (= 3.86 in this study). Therefore, we consider any particles with a measured aspect ratio from tilted SEM images greater than 3.86 as biased, and we forced them to 3.86 due to the extremely flat shape of the particles. To retrieve the correct aspect ratio of semisolid and solid particles, we applied the same conversion on tilted aspect ratios suggested by Cheng et al., 2021 if tilted aspect ratios were greater than 1.3. Thus, all aspect ratio discussed throughout this manuscript is the corrected aspect ratio. Then, we applied previously proposed corrected aspect ratio thresholds to categorize the phase state of particles; these thresholds are 1.00 to 1.30, 1.30 to 1.85, and >1.85 for solid, semisolid, and liquid states, respectively (Cheng et al., 2021). These thresholds were determined based on known RH-dependent glass transition of organic materials (e.g., Suwannee River fulvic acid (SRFA)) on the same grids type (Carbon Type-B TEM grids, Ted Pella Inc) used in this study (Cheng et al., 2021). Using the same grid type should minimize the effect of changes in surface tension and wettability, which might potentially affect the contact angle and therefore the aspect ratios. Although tilted imaging can provide a direct and convenient measurement of particle phase state, potential differences of temperature and RH inside the ESEM chamber and those at the sampling site might cause deformation of the particles.

## 3 Results and Discussions

### 3.1 Transport patterns and air mass sources

In this section, we discuss the influence of varying plume ages, transport patterns, and air mass sources on the physicochemical properties of FT aerosols. We used FLEXPART retroplumes to examine transport patterns for the Pico 2017 samples up to 20 days upwind (Fig. 1) and FLEXPART CO tracer to estimate plume ages and the relative contribution of anthropogenic and wildfire emissions (Fig. S5). Transport patterns and CO source contributions for all 2014, 2015, and 2017 samples are listed in Table 1. The plume ages and relative contributions from anthropogenic and biomass burning emissions can reveal air mass sources, types, and transport patterns. Although they do not directly reflect aerosol sources and ages, they are still good indicators to help interpret observed aerosol properties, especially in the comparisons across different samples. We also analyzed the vertical distribution of air masses residence time (Fig. S6 for S3-1 to S4-4 and Fig. S7b, S7d, S7f, S7h, S7j, and S7i for S1 to S6) and the temperature and RH during transport for a better understanding of ambient conditions that are critical for aerosol property transition. Figure 1a shows that sample S3-1 was collected during a 6-day period of recirculation over the North Atlantic Ocean. About 80 % of S3-1 air masses were contributed from anthropogenic emissions from North America based on CO sources contribution, and it was also expected to be influenced by ocean sea sprays due to its long residence time over the ocean and air mass contributions in the marine boundary layer (Fig. S6a). All the other samples (Fig. 1b to 1h) were dominated by air masses from North America but differed in transport pathways, travel distances, and impact from anthropogenic and wildfire emissions. Although samples S3-2 and S3-3 were collected on the same day, the CO contributions of sources were different, with greater contributions from North American anthropogenic sources for S3-2 (~47%) compared to S3-3 (~31%) (Table 1). Moreover, retroplumes also show that S3-3 received larger air masses from Central America than S3-2. Air masses of S3-4 were transported across the United States (contribution from anthropogenic CO sources was ~21 %) and then combined with outflow from Asia (contribution from anthropogenic CO sources was ~63 %). Air masses of S4-1, S4-3, and S4-4 were transported across the United States and received air masses from Canada, and the transport period from North America to OMP was approximately 7, 5, and 5 days, respectively. Air masses of S4-2 were transported from the Arctic and then transported across Canada, and the transport period for the S4-2 air masses from Canada to OMP was roughly 5 days. For S4-1 to S4-4, wildfire emissions also had significant CO contributions (~21 %, ~35 %, ~50 %, and ~44 %, respectively) (Table 1 and Fig. S5). The estimated average plume ages are listed in Table 1. As shown in Table 1, all samples have been highly aged during long-range transport and have similar aging times (~13 days). Sample S4-4 has the shortest aging time (~11 days), and the samples S4-1 and S4-2 have the longest aging time (~14 days).

We also performed the same FLEXPART analyses on Pico 2014 and 2015 samples (SA1 to SA3 and S1 to S6) to trace the retroplumes of air masses and sources of these air masses, and the results are listed in Table 1. FLEXPART simulation results for Pico 2014 samples have been discussed in Lata et al., 2021. Briefly, air masses for SA1 and SA2 originated in Eastern U.S.

and recirculated over the North Atlantic Ocean, and air mass for SA3 originated in Canada and the U.S. and recirculated over the North Atlantic Ocean. The average atmospheric aging time for SA1, SA2, and SA3 was ~16 days. Based on the CO tracer analysis, the major CO sources for SA1 were anthropogenic emissions in North America (~49 %), anthropogenic emissions in South America (~8 %), and wildfires in North America (~19 %). For SA2, the major CO sources were North American anthropogenic emissions (~42 %), African anthropogenic emissions (~16 %), and North American wildfires (~31 %). For SA3, anthropogenic (~49 %) and wildfire (~49 %) emissions in North America were the two major CO contributors (Lata et al., 2021). Like Pico 2014 and Pico 2017 samples, all Pico 2015 samples originated from North America and then recirculated over the North Atlantic Ocean (Fig. S7). Based on the CO tracer simulations (Fig. S8), the major source of CO for sample S2 was anthropogenic emissions in North America (~84 %), and S1, S3, S5, and S6 were influenced by both anthropogenic and wildfires CO emissions in North America (~56 %, ~79 %, ~38 %, and ~59 % for anthropogenic CO sources, and ~42 %, ~19 %, ~53 %, and ~25 % for wildfires CO sources, respectively). Wildfire emission in North America is the major CO source of S4 (~91 %). The average aging time for Pico 2015 samples varied between ~10 and ~17 days.

### 3.2 Chemical characterization of the OMP samples

### 3.2.1 Chemically-resolved size distribution of individual particle

Figure 2 shows the particle size distribution and the total particle concentration based on SMPS measurements at OMP, and CO tracer concentrations in the air masses that arrived at OMP as retrieved from FLAXPART simulations (5 July 2017 to 21 July 2017). Mobility diameter ranged from 30 nm to 500 nm, and the mode was around 60±22 nm (Fig. 2a). The total particle concentration was around 279±114 # cm$^{-3}$. The size range, size mode, and particle concentration were comparable to those found in previous studies for FT particles (10-1000 nm, <100 nm, $10^1$ to $10^4$ # cm$^{-3}$, respectively) (Igel et al., 2017; Rose et al., 2017; Sanchez et al., 2018; Schmeissner et al., 2011; Sun et al., 2021; Venzac et al., 2009; Zhao et al., 2020). Figure 2b shows that the total particle concentrations positively correlate with the CO tracer concentrations from July 5[th] to July 12[th] and from July 18[th] to 21[st], suggesting the major sources of particles during these periods might be anthropogenic and wildfire emissions. On the other hand, particle concentrations between late July 12[th] and 17[th] were above 279 # cm$^{-3}$, while the CO tracer level was relatively low (<10 ppbv) compared to other days, which might indicate additional sources of particles (e.g., sea spray and dust).

Figure 3 shows chemically-resolved size distributions inferred from the CCSEM-EDX data. The number fraction of each particle class in each sample is listed in Table S2. Tilted transmission electron microscopy (tilted angle 70°) images show that inorganic inclusions (e.g., sea salt, nitrate, sulfate, dust) are internally mixed and coated by organics (Fig. S9). Figure 3a shows the average number fraction of different particle types in each sample, and Fig. 3b to 3i show chemically-specific normalized particle size distributions. Overall, the carbonaceous aerosols (OC+CNO) have the highest number fraction in samples S3-2 to S4-4 (~42 to 69 %) and the second-highest in sample S4-1 (~30 %). Moreover, sea salt and sea salt with sulfate particles

are a significant fraction in each sample (~8.8 to 31.59 %, and ~5.2 to 31.5 %, respectively). Sea salt particles were typically from marine sprays since air masses were transported over the North Atlantic Ocean. Sea salt with sulfate particles with area equivalent diameters greater than 0.6 μm have been shown to be a product of aqueous phase processing (i.e., fog and cloud processing) (Ervens et al., 2011; Kim et al., 2019; Lee et al., 2011, 2012; Zhou et al., 2019), and those with area equivalent diameter less than 0.6 μm might have been generated from marine sources (Sorooshian et al., 2007; Yu et al., 2005). Sea salt and sea salt with sulfate particles dominated (~28.2 % and ~31.5 %, respectively) sample S3-1, with a smaller fraction of organic particles (OC and CNO, ~6.3 % and ~23.4 %, respectively) than in other samples. This result is supported by the FLEXPART retroplumes, which suggested a 6- to 7-day long recirculation over the North Atlantic Ocean, making marine spray particles have a more considerable impact. Although sample S3-2 was collected 2 hours before S3-3, their chemical compositions differed. S3-2 has more CNO and less OC and sea salt particles than S3-3, which might be due to a larger contribution from Asia and a smaller contribution from South America and the Pacific Ocean than S3-3 (Table 1 and Fig. 1b and 1c). Details of the chemically-resolved size distribution for Pico 2014 samples have been discussed in Lata et al., 2021. Briefly, all samples (SA1, SA2, and SA3) are dominated by carbonaceous particles (~68 %, ~57 %, and ~67 % by number, respectively). Besides carbonaceous particles, there are high fractions of CNOS particles in SA1 and SA3 (~14 % and ~23 %, respectively), suggesting potential cloud processing. ~11 % of particles in SA1 are dust, which might have originated from Africa. The fraction of sea salt particles is high in SA2 (~23 %) due to the longer recirculation time over the North Arctic Ocean (~15 days) (Lata et al., 2021). For Pico 2015 samples, carbonaceous particles dominate S1 and S3 to S6 (~62 %, ~63 %, ~80 %, ~52 %, and ~61 %, respectively). Sulfate (CNOS and sea salt with sulfate) particles are also abundant in all samples (~18 to 34 %), suggesting that these particles were possibly involved in cloud processing (Ervens et al., 2011; Kim et al., 2019; Lee et al., 2011, 2012; Zhou et al., 2019). Sample S2 and S5 have high dust contributions (~16 % and ~14 %, respectively) due to air masses coming from Africa. In oncoming work, we will discuss the particles' chemically-resolved size distributions in more detail.

### 3.2.2 Chemical imaging of individual particle

Figure 4 shows STXM-NEXAFS Carbon K-edge chemical speciation maps and spectra for four typical particle mixing states of OC (green), IN (cyan), and EC (red) found in S3-3 and S4-2, which are (a) organic particle (green), (b) EC core (red) and coated by OC (green), (c) internally mixed EC (red) and IN (cyan) coated by OC (green), and (d) IN (cyan) coated by OC (green). The STXM-NEXAFS spectra for these four types of particles show that one of the major components in these particles is OC since the intensity of R-COOH (288.5 eV) and R(C=O)/C−OH (286.5 eV) peaks are more intense. However, particles also contain inorganic components. Moreover, STXM images (see Fig. 4) indicate that particles are internally mixed and coated by organic species, suggesting our samples might be highly aged during transport in the FT (China et al., 2015; Motos et al., 2020). This observation is consistent with our tilted TEM images showing that EC and IN inclusions were internally mixed with organics (Fig. S9).

The particle chemically-resolved size distributions for seven samples (SA1-SA3, S3, S5, S3-3, and S4-2) analyzed with STXM-NEXAFS are shown in Fig. S10. In the S3-3 and S4-2 samples, OCIN particles are dominant (~87.8 % and ~98.8 %, respectively), and there is only a very small fraction of OC-rich particles (~5.2 % and ~1.2 %, respectively). Based on the CCSEM-EDX measurements, most IN are sea salt, which might have originated from the marine boundary layer while the air mass crossed the Atlantic Ocean. Moreover, we did not find any EC-rich, IN-rich, and OCEC in either sample, but we found a small portion of OCINEC (~7.0 %) in S3-3. For Pico 2014 samples, OCIN particles are also dominant (~83 %, ~64 %, and ~77 % for SA1, SA2, and SA3, respectively) (Lata et al., 2021). For SA1 and SA3, these IN might come from sulfate generated from cloud processing, and for SA2, these IN might come from sea sprays (Lata et al., 2021). For Pico 2015 samples, S3 is dominated by OC (~69 %). S5 also has a significant fraction of OC (~22 %), but OCIN and OCINEC are more dominating (~35 % and ~38 %, respectively). The EC might originate from wildfire (see Sect. 3.1), and IN might come from sea spray and cloud processing (see Sect. 3.1 and Sect. 3.2.1).

### 3.3 Phase state of particles

### 3.3.1 Phase state of particles at the OMP site

As discussed in Sect. 2.4, we used tilted view SEM images to determine the particles' phase state at OPM. Figure 5 shows violin plots of the 'corrected' aspect ratio (left) and representative tilted images (right) for Pico 2014 (a to c), Pico 2015 (d to i), and Pico 2017 (j to q). The number fraction of particles in each phase state for Pico 2014, 2015, and 2017 samples are listed in Table 1. As shown in Fig. 5, all samples are dominated by liquid particles (ranges from ~47 % to ~99 % by number) at the time of sample collection during summertime. We also observed considerable fractions of semisolid (~1 to 42 %) and solid particles (~0 to 30 %) for specific events, suggesting that the abundance of liquid, semisolid and solid particles in FT depend on atmospheric transport events and particle sources. A potential explanation for the high fraction of liquid particles is that our samples are internally mixed with inorganic species (e.g., sea salt and sulfate) based on the tilted TEM images, CCSEM-EDX, and STXM-NEXAFS measurements (see Sect. 3.2). Previous laboratory studies have found that at the same condition, the viscosity of a homogeneous internally mixed particle of organics and inorganics (e.g., nitrate, sulfate, and sodium salt) decreases with increasing fraction of inorganic (Dette and Koop, 2015; Power et al., 2013; Rovelli et al., 2019; Saukko et al., 2012; Schill and Tolbert, 2014; Song et al., 2021; Wang et al., 2015) since these inorganics are more hygroscopic and increasing the overall hygroscopicity of internally mixed particles. Besides, Richards et al., 2020 have reported that divalent ions (e.g., $Mg^{2+}$ and $Ca^{2+}$) can increase aerosol viscosity due to ion-molecule interactions. Although our analytical technique cannot identify the chemical formula involving these divalent ions, this phenomenon might not be critical for our samples because we found only minor fractions of Mg and Ca. Moreover, limited field studies have investigated the influence of inorganic inclusions on the viscosity and/or phase state of ambient organic aerosols and found similar results as laboratory studies (Liu et al., 2019; Slade et al., 2019). Slade et al., 2019 performed field measurements at the University of Houston Mobile Air Quality Laboratory and found that aerosols are less viscous and more liquid state aerosols during daytime than at

night due to higher amounts of inorganic sulfate. Liu et al., 2019 have found that particles in Shenzhen, China, from 17 April to 11 May 2017, were abundant in the liquid state (43±6 %) due to high inorganic mass fraction in particles (62.6±12.4 % of dry particles, on average). Thus, these laboratory and field studies support our findings since our samples have abundant inorganic species inclusions such as sea salt and sulfate (see Sect. 3.2), and the percentage of elements associated with these species (Na, Mg, Al, P, S, and K) are ranged from ~1.2 % to 15.9 % (see Table 1).

Additional to the inorganic species internally mixed with organic species, another potential explanation for the abundance of liquid particles in our samples might be that our samples were highly aged during long-range transport since their average aging times were more than 10 days. Aging processes can reduce the particles' viscosity due to photodegradation and photobleaching (Koop et al., 2011; Malecha and Nizkorodov, 2016; Pajunoja et al., 2016; Pan et al., 2009; Sumlin et al., 2017). A previous study at Pico investigated the molecular composition of water-soluble organic matter and found compounds with low oxygenation (Dzepina et al., 2015). The authors hypothesized that aqueous-phase processing (cloud processing) and fragmentation (photolysis) might be responsible for the selective removal of some fraction of highly aged polar compounds. Similarly, another study at the site highlighted organic components (after removing the inorganic salts) with lower O/C ratio, and particles were in the solid state during the long-range transport (Schum et al., 2018). These results suggest that apart from environmental factors, the inorganic components, the molecular weight of organic compounds, and the O/C ratio (or aging time) all affect the phase state of internally mixed particles. Thus, we hypothesize that a substantial fraction of solid and semisolid particles might be less oxidized and less prone to be removed via aqueous-phase processes than liquid particles in the FT during transport.

Besides these two potential explanations, many aspects can still affect the phase state of particles. Particles can transit from solid to semisolid to liquid state when RH and/or temperature increase (Koop et al., 2011). Thus, these particles might transit to different phase states if the ambient conditions change. For example, measured RH at OMP was highest during the S2 and S3 sample collection periods (61.3±2.4 % and 67.3±2.3 %, respectively) and lowest during the S4-2 and S4-3 collection periods (6.6±0.3 % and 9 %, respectively). The lower RH at OMP during S4-2 and S4-3 collection periods might help explain the observation of more abundant solid state particles in S4-2 and S4-3 rather than S2 and S3.

Moreover, Cheng et al., 2015, Petters and Kasparoglu, 2020, and Kaluarachchi et al., 2022 have shown that particle size also affects the particles' viscosity. This appears to be the case for some samples when comparing the aspect ratio distribution for the Pico 2015 particles collected on stage 3 (left violin plots, 50 % cut-off size is >0.15 μm) with those from stage 4 (right violin plots, 50 % cut-off size is >0.05 μm) in Fig. 5d to 5i. For samples S1, S2, S4, and S6, particles from stage 3 have lower mode and mean aspect ratio than those from stage 4, indicating that larger particles have higher fractions of more viscous particles than smaller particles. However, the aspect ratio distributions for particles collected on stage 3 in samples S3 and S5 have higher modes and mean values than those on stage 4, suggesting a higher fraction of less viscous particles in S3 and S5

on stage 3. Due to these inconsistent observations and the limited number of samples, we cannot draw clear conclusions regarding the size dependence of particle viscosity; this important aspect should be the objective of future studies.

In addition, we observed two different types of aspect ratio distributions: (a) narrow distribution with mean aspect ratios below 4 and a smaller fraction of particles (< 35 %) with aspect ratios greater than 4 (standard deviations of aspect ratio were ranging from 0.9 to 2.1) (Fig. 5a, 5d, 5e, and 5i to o), and (b) broad distribution with a larger fraction of particles (> 40 %) with aspect ratios greater than 4 (standard deviation of aspect ratio were ranging from 1.4 to 2.4) (Fig. 5b, 5c, 5f, 5g, 5h, 5p, and 5q). The correlation analyses of aspect ratio against the FLEXPART CO tracer suggest that sources of samples that have the type (b)

distribution usually have a high CO contribution from wildfires (>30 %). There are two exceptions, S4-2 from Pico 2017 and S1 from Pico 2015, which received wildfire CO contributions greater than 30 % (~35 % and ~42 %, respectively) but have type (a) distribution shape. For S4-2, a possible reason is that the volatile and less viscous species in particles collected on the TEM grid have already evaporated and left these tiny residuals around those big particles (see Fig. 5f right panel) due to difference in temperature, RH, and pressure between OMP and the SEM chamber, a phenomenon which has not been observed

in other samples. Hence, those particles remaining on the substrate in S4-2 have a higher viscosity than the original particles. For S1, the altitude of the center of air mass during transport was lower than other samples (Fig. S7b), which might make S1 have more considerable impacts from PBL. To test the hypothesis that source and FT aerosols' viscosity are correlated, we applied a paired-difference t-test ($\alpha=0.05$) of the number fraction of particles that have aspect ratios greater than 4 (see Table 1) and the contribution of wildfires CO source based on the FLEXPART CO tracer simulation (see Table 1), and the results

suggest that these two are positively correlated (p= 0.0186). It should be noticed that we only have a limited number of samples (17 samples) in this study. Thus, it is necessary to perform more studies to improve our knowledge of the influence of sources on FT particles' phase state and viscosity.

Moreover, we also observed a considerable fraction of semisolid and solid particles in samples with high wildfire CO

contribution (~1 to 29 % and ~0 to 16 %, respectively), suggesting wildfire-influenced plumes can also emit high viscous particles. The presence of high viscosity organic particles in wildfire-influenced plumes in FT after long-range transport is in accordance with a previous study at Pico (Schum et al., 2018). Therefore, our results suggest that the viscosity of FT particles depends on their source, and wildfire-influenced plume results in particles with a wide range of viscosities in the FT. To date, the knowledge of the phase state and/or viscosity of wildfire aerosols is limited. Many studies have reported observations of

tar balls, a solid spherical organic aerosol, in biomass emissions (Adachi et al., 2019; Adachi and Buseck, 2011; China et al., 2013; Hoffer et al., 2017; Posfai et al., 2004). DeRieux et al., 2018 predicted the viscosity of biomass burning particles using their chemical composition, and they found that the viscosity of biomass burning particles varied between $10^{-2}$ Pa·s and $10^{9}$ Pa·s depending on the RH. Liu et al., 2021 found that ambient and lab-generated biomass burning particles are in a non-solid state at RH between 20 % and 50 %. Li et al., 2020 predicted the viscosity of ambient biomass burning organic aerosols from

volatility distribution and found that it varies between $10^{-2}$ Pa·s (liquid state) to above $10^{12}$ Pa·s (solid state) in Athens (Greece)

and $10^{-2}$ Pa·s (liquid state) to $\sim 10^9$ Pa·s in Mexico City (Mexico). Therefore, these studies support our finding that biomass burning can emit particles with a broad viscosity distribution. This finding underlines the importance of constraining source-specific contributions to determine the phase state of FT particles, which is still not well documented.

Besides tilted imaging, we also investigated the phase state of particles utilizing STXM-NEXAFS measurements to retrieve the TCA. Typically, particles with the same area equivalent diameter but higher TCA are more viscous (more solid-like) since they are less flat in shape (Fraund et al., 2020; Tomlin et al., 2020). However, if a particle has a solid inorganic core (e.g., dust or soot) and organic coating, its TCA will be low, indicating a low thickness. In this case, it is difficult to probe the phase state of the organic coating; for these cases, tilted imaging provides better estimates since we can directly observe the thickness of
entire particle. Figure 6 shows the TCA as a function of the area equivalent diameter of the impacted particles for samples from Pico 2014, Pico 2015, and Pico 2017 (left panel) and a histogram plot of OVF to investigate the contribution of inorganic inclusions in phase state of particles. Symbols are colored by their OVF. Shaded areas represent regions of different phase states (liquid: blue, semisolid: green, and solid: red), with the boundaries of each region based on measurements of field and lab-generated organic particles reported in O'Brien et al., 2014. As shown in Fig. 6, TCA values for our samples are very low,
and more than 90 % of the particles in each sample are in the liquid state, consistent with our tilted imaging measurements. Except for S3, all samples have OVF around 0.20±0.22, suggesting these carbonaceous particles contain other elements.

### 3.3.2 Phase state of particles during long-range transport

Due to changing atmospheric conditions, the phase state of aerosols during transport might be more variable than that at the sampling site. To investigate the phase state of particles during transport, we estimated the expected phase state of the samples
indirectly by calculating their glass transition temperature ($T_g$), which is the transition temperature between solid and semisolid states (Koop et al., 2011). To calculate $T_g$ of a mixture (e.g., internally mixed organics and inorganics), we need to know the properties of each component (e.g., mass fractions, density, hygroscopicity, and dry $T_g$). Due to limited information on inorganic species' type and volume fraction in our samples, we can only predict RH-dependent $T_g$ values of organic particles ($T_{g,org}$) (Koop et al., 2011). Typically, organic particles maintain solid states when $T_{g,org}/T \geq 1$, semisolid state when $1 > T_{g,org}$
$/T \geq 0.8$, and liquid state when $T_{g,org}/T < 0.8$ (Schmedding et al., 2020; Shiraiwa et al., 2017). We calculated expected $T_{g,org}$ during the air mass transport using the temperature and RH extracted from the GFS along the FLEXPART simulated path. We used the density ($\rho_{org}$), hygroscopicity ($\kappa_{org}$), and dry glass transition temperature ($T_{g,org,0}$) of organic particles as reported by Schum et al., 2018 (see SI Sect. S2) since we do not have molecular compositions for our samples and Schum et al., 2018's samples were also collected at OMP during the same seasonal period (June and July).

Predicted $T_{g,org}$ based on FLEXPART for each Pico 2017 sample during transport are shown in Fig. S11. The estimates only up to 5 days before air masses arrived at OMP to avoid increased uncertainties associated with possible meteorological conditions due to the spread of the air masses (see Fig. S6). Predicted $T_{g,org}$ to $T$ ratio as a function of temperature and relative

humidity are shown in Fig. S12. The extracted ambient temperature for all samples was relatively stable during transport and slightly increased when the air masses approached OMP. On the other hand, ambient RH was more variable from grid cell to cell. The RH decreased when the air mass reached OMP. Therefore, ambient RH and dry $T_g$ (see SI Sect. S2) yield a wide range of $T_{g,org}$ values (191.1K – 329.1K) during transport (Fig. S11). Overall, the predicted average $T_{g,org}$ values in all samples exceed the ambient temperature extracted from FLEXPART during most of the transport periods (Fig. S11), implying that organic particles would likely be solid in the FT. These results are consistent with a previous study at the site (Schum et al., 2018) and other modeling studies (Li et al., 2021; Schmedding et al., 2020; Shiraiwa et al., 2017; Shrivastava et al., 2017) that organic particles in the FT are likely in solid and semisolid states. We also calculated $T_{g,org}$ of organic particles at OMP using meteorological data collected at OMP (Table S2), and it also shows that the organics in our samples should have been in solid state at the time they were sampled at OMP since the predicted $T_{g,org}$ is higher than the measured temperature. Moreover, $T_{g,org}$ at OMP calculated using FLEXPART data is close to that calculated using the local meteorological data (difference <4.4 %), and the difference between FLEXPART simulated temperature and RH at the site and those measured at OMP is not significant (less than 4 °C and RH varies between ~3.5 % to ~12 %), which indicate that the FLEXPART data can be used to predict the phase state of organic particles during long-range transport.

In addition to the potential difference in the molecular formula of organics and ambient conditions between our study and Schum et al., 2018, our offline phase state measurements show that the samples collected at OMP might have lower $T_g$ than the theoretically predicted $T_{g,org}$. This might be due to the fact that our sampled particles were internally mixed with hygroscopic inorganic species (e.g., sea salt and sulfate, see Sect. 3.2), which are expected to decrease the viscosity and the $T_g$ of the particles (Dette and Koop, 2015; Power et al., 2013; Rovelli et al., 2019; Saukko et al., 2012; Schill and Tolbert, 2014; Song et al., 2021; Wang et al., 2015). Moreover, in the calculation of $T_{g,org}$, we assumed the CCN-derived $\kappa_{org}$ to be 0.12 (Schum et al., 2018), which might be lower than that of ambient internally mixed particles (Ching et al., 2019; Kristensen et al., 2016; Pringle et al., 2010; Schulze et al., 2020). The underestimation of $\kappa$ would lead to an overestimation of $T_g$. Thus, our results suggest that estimating the phase state of particles without considering the mixing state of FT particles might not accurately predict their viscosity and $T_g$ because the presence of hygroscopic inorganic inclusions (e.g., sea salt and sulfate) can reduce the viscosity of FT particles at the RH and T values encountered in the FT during transport. Moreover, our results highlight the essential roles of FT sites such as OMP for studying the phase state and viscosity of FT particles since these sites provide unique opportunities for comprehensively understanding FT aerosols' chemical composition, physical properties, and hygroscopicity. This information is necessary to predict the particles' phase state in the FT more accurately, which is critical for predicting their effects on climate.

## 4. Conclusions

The phase state and viscosity of free tropospheric particles are important properties for estimating the climatic effects of aerosols. However, the current understanding of phase state and viscosity is limited, especially for aerosol in the remote free troposphere. In this study, we analyzed the chemical properties and phase state of single particles collected in the free troposphere at the OMP, in the Azores, over three different years. We combined the single-particle analysis with FLEXPART analysis to estimate the particles' phase state during transport. We utilized multi-modal micro-spectroscopy techniques

(CCSEM-EDX, tilted view imaging combined ESEM, and STXM-NEXAFS) to probe the chemical composition, phase state, and mixing state of long-range transported free-tropospheric particles. The source and air mass transport trajectories and ages were also determined using FLEXPART. FLEXPART CO tracer simulations showed that North American wildfire emissions primarily influenced SA3, S1, S3, S6, S4-2, S4-3, and S4-4, while the rest of the samples were largely influenced by anthropogenic sources. CCSEM-EDX analysis indicates that carbonaceous aerosols are the dominant type in all samples.

Carbonaceous sulfate and sea salt with sulfate particles are also present, probably due to the contribution of marine sprays and cloud processing. The CCSEM-EDX analysis and FLEXPART simulations revealed that transport patterns could affect particle chemical composition. The offline single particle phase state analysis showed that the major fraction of particles was liquid (~47 to 99 %). However, there were also a considerable fraction of solid (~0 to 30 % by number) and semisolid (~0.1 to 42 %) particles, and their relative abundance depends on the transport pattern and particle sources. We hypothesize that the high

abundance of liquid particles is due to the presence of inorganic inclusions (e.g., sulfate and Na salt) and the long atmospheric aging times (>10 days), which can reduce the viscosity of organics due to photodegradation and photobleaching. Moreover, we found that samples with a significant influence of wildfire emissions (wildfire CO contribution >30 %) have a broad viscosity distribution than those collected during events dominated by anthropogenic emissions. This finding underscores the need to study further the influence of emission sources on the viscosity of FT particles. Our results suggest that neglecting the

contribution of internally mixed inorganic species while calculating the particle hygroscopicity properties of organic aerosols might overestimate the viscosities of internally mixed particles. Indeed, the inorganic inclusions reduced the calculated particles' viscosity, implying that the particle's probability of remaining liquid in the FT is enhanced. Moreover, the fraction and chemical composition of inorganic inclusions may further influence the phase state variations. These effects have been previously shown in laboratory studies by measuring the viscosity of organic and inorganic mixtures (Dette and Koop, 2015;

Power et al., 2013; Richards et al., 2020; Rovelli et al., 2019; Saukko et al., 2012; Schill and Tolbert, 2014; Song et al., 2021; Wang et al., 2015) and in ambient samples (Liu et al., 2019; Slade et al., 2019). Moreover, the effects of internally mixed inorganics on the particle's viscosity are not considered in current climate models (Li et al., 2021; Rasool et al., 2021; Schmedding et al., 2020; Shiraiwa et al., 2017; Shrivastava et al., 2022). These liquid particles are expected to be less stable and have faster heterogeneous reaction rates, be more reactive, and be quickly photodegraded in the FT (Berkemeier et al.,

2016; Kuwata and Martin, 2012; Lienhard et al., 2015; Liu et al., 2018a; Marshall et al., 2016; Pöschl and Shiraiwa, 2015b; Renbaum-Wolff et al., 2013). This study assesses the phase state of internally mixed FT particles at the time of sample

collection, and highlights the importance of accounting for inorganic inclusions to evaluate the phase state of internally mixed particles. Our results might not fully represent the phase state of FT particles during transport due to differences in ambient temperature and RH. Moreover, the aspect ratio thresholds used to determine the particles' phase states are based on limited

standards. Future studies should focus on improving the aspect ratio thresholds by using more standards with known viscosities and determining the viscosity of internally mixed individual particles as a function of temperature and RH.

**Supplement.** The supplement related to this article is available online at:

**Author Contribution.** ZC, CM, and SC designed the study. BW, SH, LM, and CM collected samples and performed online measurements. ZC, MM, NNL, RB, and SC performed microscopy experiments and analysis; ZC, NNL, MAM, and SC performed STXM/NEXAFS experiments and analysis. BZ performed FLEXPART simulation. ZC wrote the first manuscript draft. All authors reviewed and edited the manuscript.

**Data Availability.** All the measurement data are provided in the supplement

**Competing Interests.** The authors declare that they have no conflict of interest.

**Acknowledgments.** A portion of this research was performed on a project award (10.46936/lser.proj.2019.50835/60000118)
from the Environmental Molecular Sciences Laboratory, a DOE Office of Science User Facility sponsored by the Biological and Environmental Research program under Contract No. DE-AC05-76RL01830. STXM/NEXAFS analysis at beamline 5.3.2.2 of the Advanced Light Source at Lawrence Berkeley National Laboratory is supported by the Director, Office of Science, Office of Basic Energy Sciences of the U.S. Department of Energy under Contract No. DE-AC02- 05CH11231. We acknowledge funding from the U.S. Department of Energy, Office of Science (BER), Atmospheric System Research. The
sample collection at Pico Mountain Observatory was supported with funding from NSF (AGS-1110059), DOE (DE-SC0006941), and NASA's Earth and Space Science Graduate Fellowships (NNX12AN97H, and NNX13AN68H). Logistic support for the Observatory's operation was provided by the Regional Government of the Azores through the Regional Secretary for Science and the Pico Island Natural Park. We thank Mike Dziobak, Kendra Wright, Sumit Kumar, Andrea Baccarini, Stefano Viviani, Jacques Huber, and Detlev Helmig for their support in the field. Measurements in 2017 were
funded by the German Science Foundation (Deutsche Forschungsgemeinschaft, DFG) under SI 1543/4-1, WE 2757/2-1, and HE 6770/2-1.

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

**Table 1. Summary of Pico 2014, 2015, and 2017 samples. "S3-" and "S4-" for Pico 2017 samples were collected on stage 3 (50 % cut-off size: >0.15 µm) and stage 4 (50 % cut-off size: >0.05 µm) of a four-stage cascade impactor (MPS-4G1), respectively. Pico 2014 samples were collected on stage 3, and Pico 2015 samples were collected on stages 3 and 4 of the cascade impactor. Particle percentages of different phase state were retrieved based on tilted imaging. Additional information on sampling time and conditions and fractions of different species in each sample based on CCSEM-EDX and STXM-NEXAFS is listed in Tables S1 and S2.**

| Sample ID | Air mass pattern | Contribution of source | | | | | | Average aging time (days) | % of solid particle | % of semisolid particle | % of liquid particles | #% of particle has aspect ratio ≥ 4 | Element percentage of Na, Mg, Al, P, S, and K |
|---|---|---|---|---|---|---|---|---|---|---|---|---|---|
| | | North America | Europe | Asia | South America | Africa | Wildfire | | | | | | |
| SA1 (2014) | North America | 48.9% | 8.4% | 0.0% | 13.7% | 0.3% | 28.6% | 16.4 | 29.8% | 23.1% | 47.1% | 0.0% | 5.8±6.4% |
| SA2 (2014) | North America | 41.7% | 0.2% | 0.9% | 0.1% | 26.2% | 31.0% | 16.2 | 0.0% | 1.9% | 98.1% | 84.6% | 15.9±19.9% |
| SA3 (2014) | North America | 49.0% | 0.0% | 0.4% | 0.1% | 1.1% | 49.4% | 16.0 | 1.0% | 1.0% | 98.0% | 68.6% | 9.1±7.1% |
| S1 (2015) | North America | 55.8% | 0.0% | 2.1% | 0.0% | 0.0% | 42.2% | 15.7 | 4.5% | 28.5% | 67.0% | 20.0% | 1.7±3.3% |
| S2 (2015) | North America + Africa | 83.7% | 0.0% | 7.2% | 1.7% | 3.3% | 4.1% | 12.2 | 1.0% | 5.5% | 93.5% | 29.0% | 6.1±7.2% |
| S3 (2015) | North America + Arctic | 79.2% | 0.2% | 1.8% | 0.1% | 0.0% | 18.7% | 10.5 | 0.5% | 1.5% | 98.0% | 53.5% | 2.8±7.0% |
| S4 (2015) | North America + Arctic | 8.3% | 0.1% | 0.9% | 0.0% | 0.0% | 90.6% | 12.3 | 0.0% | 1.5% | 98.5% | 90.5% | 1.9±3.5% |
| S5 (2015) | North America + Africa | 37.9% | 0.0% | 1.4% | 3.1% | 4.7% | 52.9% | 16.1 | 0.5% | 5.5% | 94.0% | 45.5% | 4.4±5.7% |

| S6 (2015) | North America | 59.3% | 0.4% | 0.0% | 10.4% | 4.7% | 25.2% | 17.1 | 1.5% | 42.0% | 56.5% | 15.0% | 3.3±4.5% |
|---|---|---|---|---|---|---|---|---|---|---|---|---|---|
| S3-1 (2017) | Recirculation over the North Atlantic Ocean | 80.3% | 0.6% | 11.3% | 2.7% | 0.0% | 5.1% | 13.3 | 8.3% | 15.5% | 76.1% | 12.5% | 2.2±1.6% |
| S3-2 (2017) | North America with contribution from Africa | 47.3% | 0.0% | 0.0% | 32.0% | 17.1% | 3.6% | 12.8 | 0.0% | 6.7% | 93.3% | 30.9% | 1.3±1.4% |
| S3-3 (2017) | North America + South America+ Africa | 31.0% | 0.0% | 0.0% | 50.2% | 16.0% | 2.9% | 12.7 | 2.0% | 4.7% | 93.3% | 35.3% | 1.2±1.8% |
| S3-4 (2017) | North America + Europe + Asia | 20.5% | 0.9% | 63.3% | 0.6% | 0.6% | 14.2% | 12.5 | 3.1% | 3.1% | 93.8% | 32.9% | 1.4±1.5% |
| S4-1 (2017) | North America | 69.7% | 0.3% | 9.4% | 0.0% | 0.0% | 20.5% | 14.0 | 11.0% | 9.0% | 80.0% | 31.5% | 1.3±1.0% |
| S4-2 (2017) | Asia + Arctic | 1.2% | 3.1% | 60.5% | 0.0% | 0.0% | 35.2% | 14.2 | 5.8% | 13.1% | 81.0% | 8.0% | 1.3±0.8% |
| S4-3 (2017) | North America | 22.4% | 0.5% | 25.1% | 0.0% | 0.0% | 52.0% | 13.1 | 10.2% | 5.6% | 84.3% | 52.3% | 1.6±1.1% |
| S4-4 (2017) | North America | 48.4% | 0.1% | 7.1% | 0.0% | 0.1% | 44.3% | 11.3 | 15.5% | 5.3% | 79.1% | 53.5% | 1.5±1.2% |

**Table 2. Acronyms and Abbreviations**

| Acronym | Definition |
|---|---|
| FT | Free Troposphere and Free Tropospheric |
| ESEM | Environmental Scanning Electron Microscope |
| PBL | Planetary boundary layer |
| OMP | The Observatory of Mount Pico |
| RH | Relative humidity |
| CCN | Cloud condensation nuclei |
| INP | Ice nucleating particles |
| Pico 2014 | Atmospheric aerosol particles collected at OMP during July 2014 |
| Pico 2015 | Atmospheric aerosol particles collected at OMP during June and July of 2015 |
| Pico 2017 | Atmospheric aerosol particles collected at OMP during June and July of 2017 |
| FLEXPART | FLEXible PARTicle Lagrangian particle dispersion model |
| SMPS | Scanning Mobility Particle Sizer |
| GFS | Global Forecast System |
| EDGAR | Emissions Database for Global Atmospheric Research |
| STXM-NEXAFS | Scanning transmission X-ray microscopy with near-edge X-ray absorption fine structure spectroscopy |
| CCSEM-EDX | Computer-controlled scanning electron microscopy with energy dispersive X-ray spectroscopy |
| OC | Organic |
| CNO | Carbonaceous with nitrogen |
| CNOS | Carbonaceous with sulfate |
| Na-rich | Sea salt |
| Na-rich with S | Sea salt with sulfate |
| ALS | Advanced Light Source |
| TCA | Total carbon absorbance |
| OVF | Organic volume fraction |
| OD | Optical density |
| EC | Elemental carbon |
| IN | Inorganic components |
| $t_{OC}$ | Thickness of organic |
| $t_{EC}$ | Thickness of elemental carbon |
| $t_{IN}$ | Thickness of inorganic |
| OC-rich | Particles have OC is greater than 96 % and both EC and IN are less than 2 % |
| EC-rich | Particles have EC is greater than 96 % and both OC and IN are less than 2 % |
| IN-rich | Particles have IN is greater than 96 % and both OC and EC are less than 2 % |
| OCEC | Particles have both OC and EC are greater than 2 %, and IN is less than 2 % |
| OCIN | Particles have both OC and IN are greater than 2 %, and EC is less than 2 % |
| OCINEC | Particles have OC, EC, and IN are all greater than 2 % |
| $W_{corrected}$ | Corrected particle width |

| | |
|---|---|
| H$_{\text{corrected}}$ | Corrected particle height |
| W$_{\text{tilted}}$ | Maximum particle length in the horizontal |
| H$_{\text{tilted}}$ | Maximum particle length in the vertical |
| SRFA | Suwannee River fulvic acid |
| TEM | Tilted transmission electron microscopy |
| $T_{\text{g}}$ | Glass transition temperature |
| $T_{\text{g,org}}$ | $T_{\text{g}}$ values of organic particles |
| $\rho_{\text{org}}$ | Density of organics |
| $\kappa_{\text{org}}$ | Hygroscopicity of organics |
| $T_{\text{g,org,0}}$ | Dry glass transition temperature of organics |

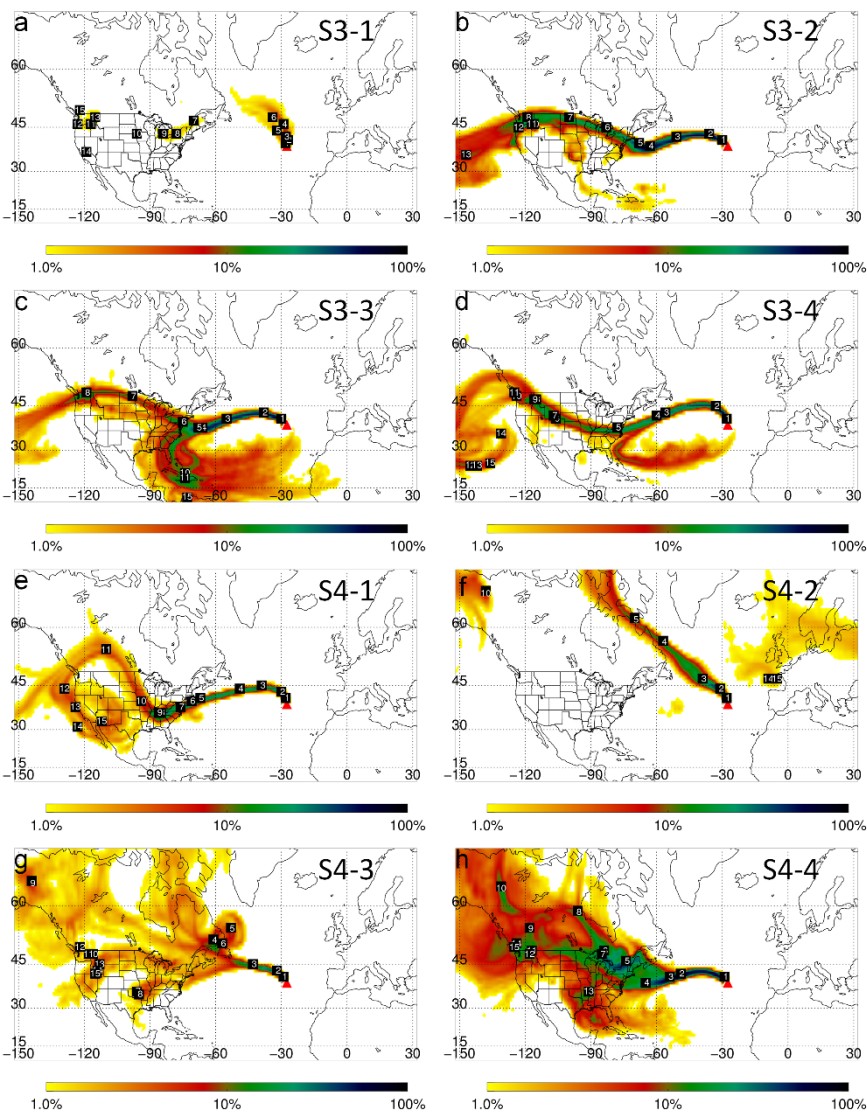

**Figure 1. Column-integrated residence time over the 20-day transport time retrieved from FLEXPART retroplumes for 2017. (a) S3-1, (b) S3-2, (c) S3-3, (d) S3-4, (e) S4-1, (f) S4-2, (g) S4-3, (h) S4-4. The vertical distribution of the retroplumes residence time at given upwind times are shown in Fig. S6. The color bars indicate the ratio of column integrated residence time to the maximal residence time at each upwind time in the logarithmic scales, and the X-axis and y-axis represent latitude and longitude, respectively. The numbers indicate locations of the highest vertically integrated residence time on a given upwind day.**

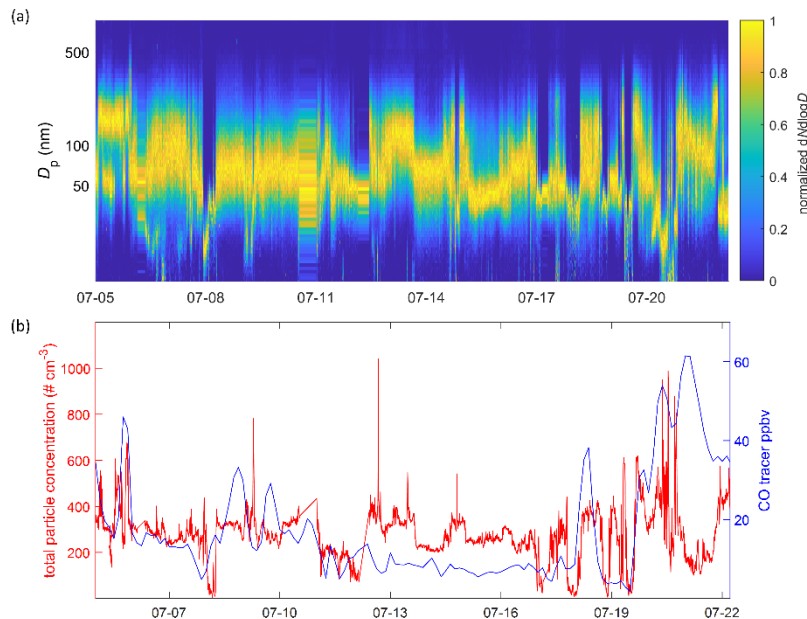

**Figure 2. (a) Normalized particle size distribution from 10 to 800 nm measured from SMPS measurements, and (b) SMPS derived total particle concentrations (left y-axis, red line) and CO tracer retrieved from FLEXPART simulations (right y-axis, blue line) from 05 to 21 July 2017.**

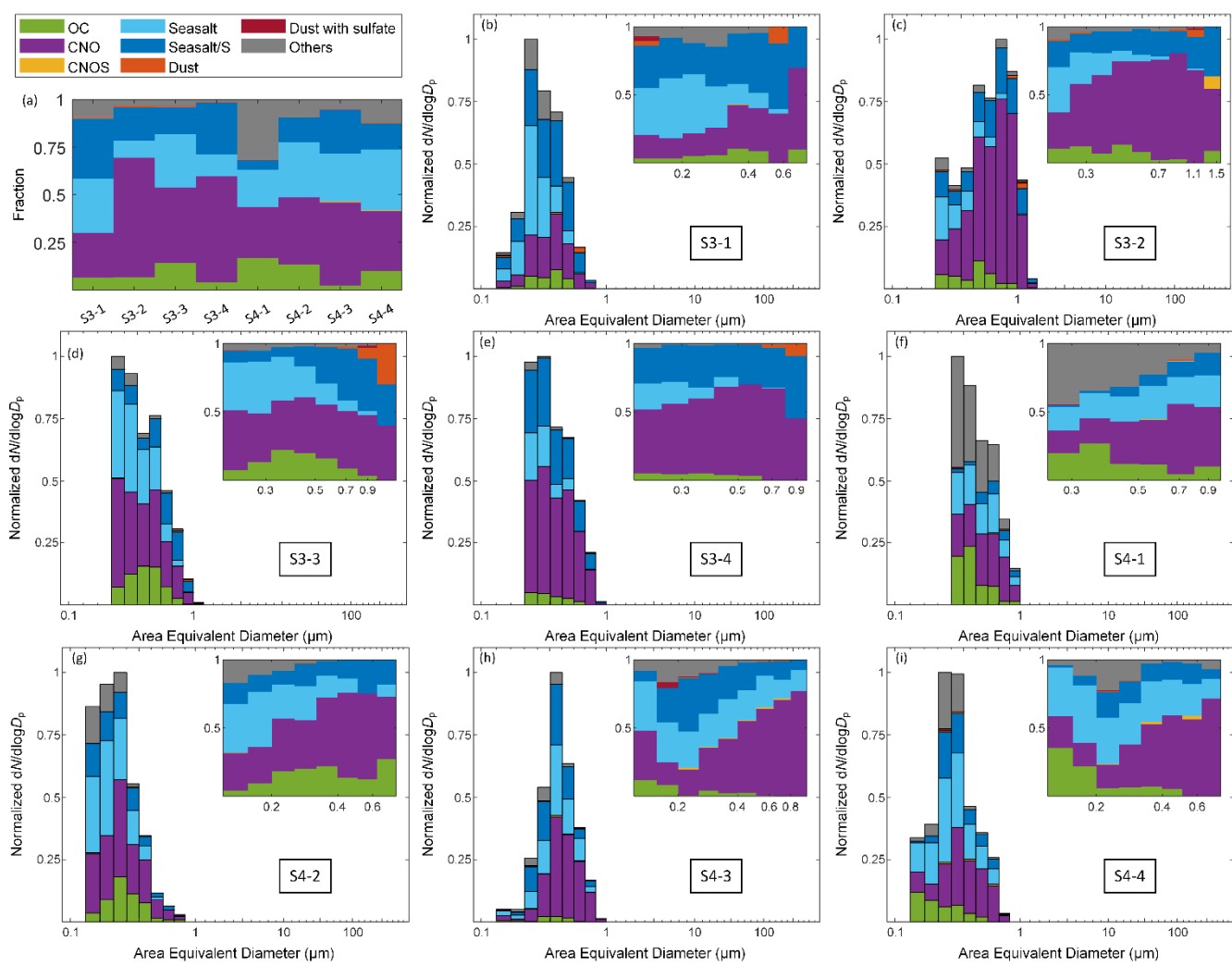

Figure 3. Chemically-resolved size distributions were inferred from the CCSEM-EDX data for 2017. (a) Fraction of different particle types for all samples. Normalized chemically-resolved size distributions of (b) S3-1, (c) S3-2, (d) S3-3, (e) S3-4, (f) S4-1, (g) S4-2, (h) S4-3, and (i) S4-4. Inserts represent the normalized number fraction of different particle types as a function of particle size.

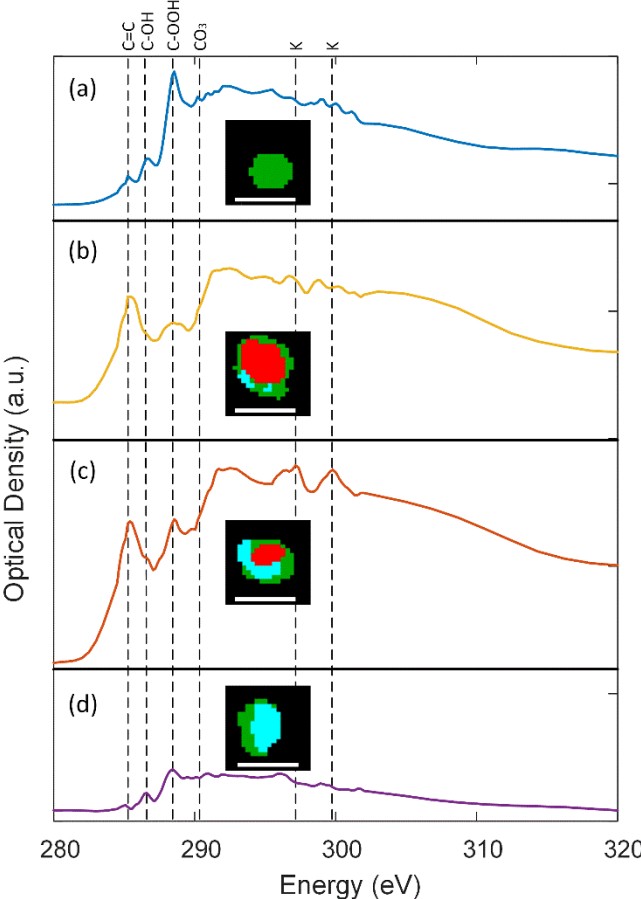

**Figure 4. Representative STXM-NEXAFS spectra of (a) organic particle (green), (b) EC core (red) coated by OC (green), (c) internally mixed elemental carbon (red), and IN (cyan) core coated by OC (green), and (d) IN (cyan) core coated by OC (green) from Pico 2017 S3-3 and S4-2 samples. White scale bars represent 500 nm.**

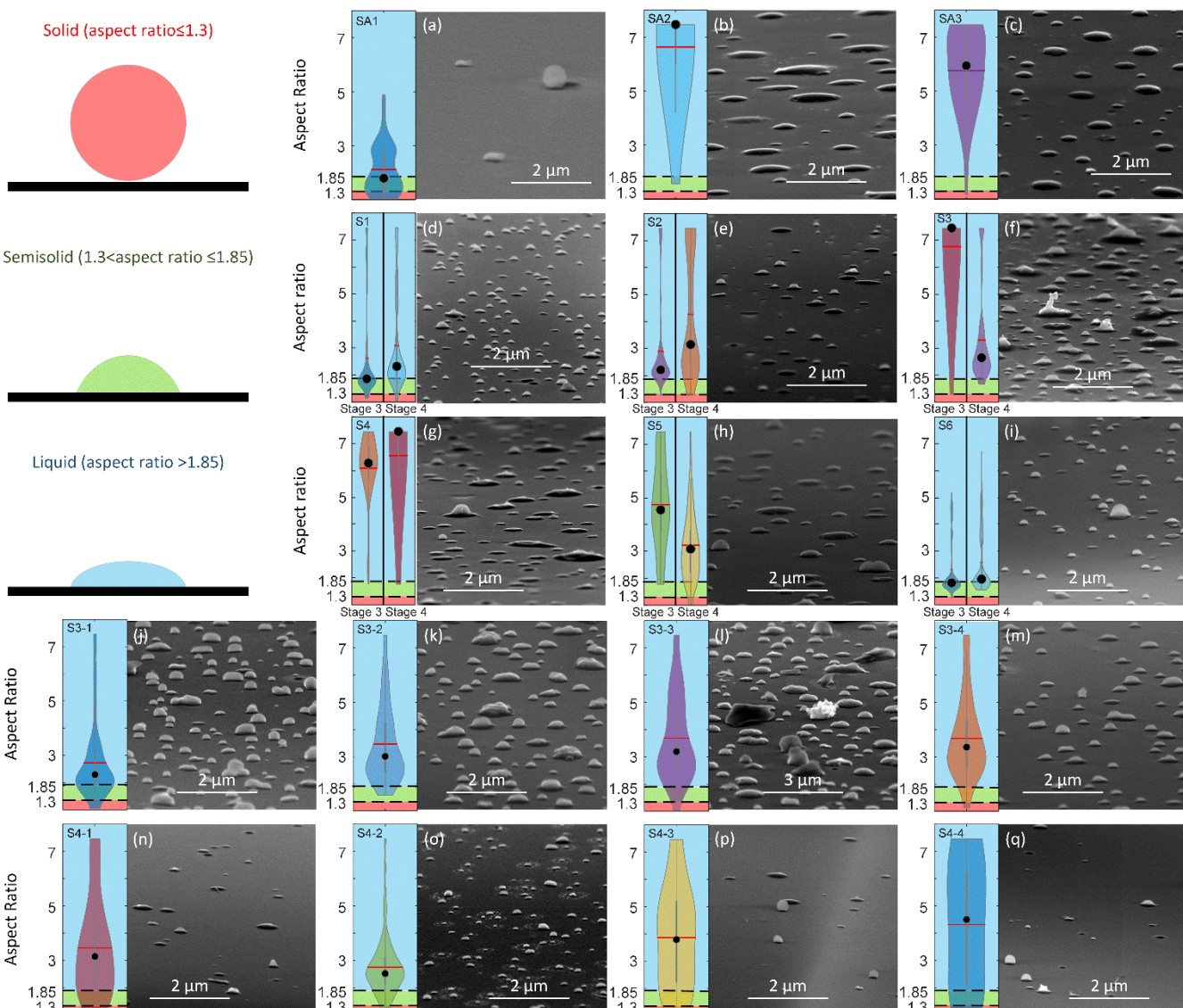

**Figure 5. Violin plots of 'corrected' aspect ratio (left) and typical tilted images (right) of Pico 2014 (a to c), Pico 2015 (d to i), and Pico 2017 (j to q). Distributions in the left panels of (d to i) are the aspect ratio of Pico 2015 particles collected on stages 3 (left) and 4 (right). The shaded region corresponds to the different phase states (red: solid state; green: semisolid state; blue: liquid state). The red lines indicate the means, and the black dots the medians.**

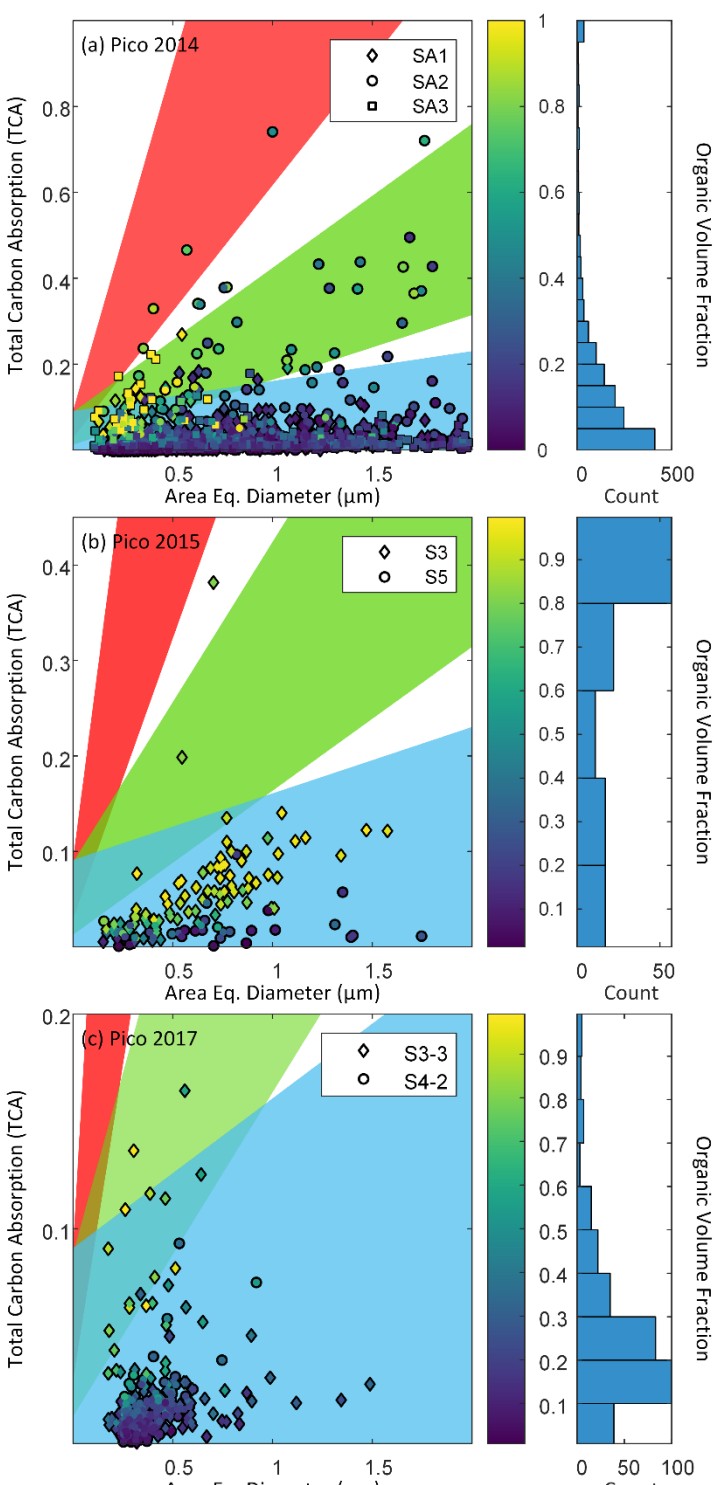

**Figure 6. Total carbon absorption (TCA) as a function of area equivalent diameter of the impacted particles (a) and histogram plot of TCA from (a) Pico 2014, (b) Pico 2015, and (c) Pico 2017. Symbols are colored by their organic volume fraction retrieved from STXM-NEXAFS measurements. Shaded areas represent different phase state regions (liquid: blue, semisolid: green, and solid: red). Side plots are histograms of the organic volume fraction.**