# Peer review of "Particles' phase state variability in the North Atlantic free troposphere during summertime determined by different atmospheric transport patterns and sources"

_Atmospheric Chemistry and Physics, 2022_

## Editor Comment (EC1)

Referee comment on "Particles' phase state variability in the North Atlantic free troposphere during summertime determined by different atmospheric transport patterns and sources" by Cheng et al. submitted to ACP

Cheng et al. made a comprehensive investigation on the sources, chemical composition, and phase state of long-range transported free tropospheric particles based on individual particle analysis using multi-modal micro-spectroscopy techniques. This study found that most particles were in the liquid state and highlighted the importance of considering the mixing state, emission source, and transport patterns of particles when estimating particle phase state in the free troposphere. Though the findings are expected, the observation data provide valuable information constraining the physiochemical properties of aerosols in the free troposphere which is important in assessing aerosol associated climate effects. I agree with the comments of the other two referees that this manuscript can be published only after a major revision as there are too many errors in the submitted version.

**Major comments:**

1. My major concern is on the Section 3.3.2, the phase state of particles during long-range transport. The authors mainly investigated the phase state of organic particles applying the temperature and RH along the air mass transport path, and found that organic particles would likely be solid in most of the times. As the particle viscosity depends significantly on RH as pointed by the other two referees, I doubt the meaningfulness investigating the phase state at each air mass path with a wide variation in RH as shown in Fig. S10. I suggest adding a figure showing the variation of predicted viscosity with RH and T (similar to the figures in (Li and Shiraiwa 2019, Petters, Kreidenweis et al. 2019)) and investigating the phase state at free troposphere-relevant conditions. In addition, the authors applied a single value for the dry glass transition temperature, which, however, would be changed due to the change in the chemical composition during the long range transport. Finally, could the authors add some discussion that based on the inorganic component types you have observed, how you expect the phase state variation of inorganic components during the long range transport at free tropospheric RH

and *T*? It would be helpful supporting the implication what you wrote in the Conclusion section that the particles in the FT probably remain liquid.

2. Mixing state plays an important role in the phase state of ambient particles; however, the authors did not mention other factors that may impact phase state significantly. Besides the influences of surface tension on aspect ratio and thus the prediction of phase state mentioned by Referee #1, the influence of particle size should be considered and discussed as well. Several studies have found that the size of particles influence the viscosity (Cheng, Su et al. 2015, Petters and Kasparoglu 2020). Did you see the difference in the phase state between the particles collected on the 3[rd] and 4[th] states of the impactor? Would the change of particle size affect the phase state during the long range transport? Secondly, the authors only mentioned the inorganic components could decrease the viscosity of internally mixed particles. They missed a recent study showing that increasing inorganic fraction can increase aerosol viscosity through cooperative ion-molecule interactions (Richards, Trobaugh et al. 2020).

**Specific comments:**

**Manuscript:**

3. I recall the comments by the other two referees that the RH in the ESEM should be clearly pointed out as the particle phase state depends significantly on RH.

4. Give the full name of "SEM" at Line 68 instead of at Line 71. Are SEM and ESEM the same?

5. Line 125. Change "Experimental" to Experiments.

6. Line 177-178. "87 for S3 and 37 for S5 for Pico 2015, and 142 and 171 particles for S3-3 and S4-2 for Pico 2017".
   These data are not same as those in Table S2 and Table S3. Please check which are correct.

7. Line 190, I don't understand what TCA is proportional to?

8. Line 245. Explain how you determined the air mass source is wildfire from "CO source contributions".

9.  Line 277-281, the data described for SA1, SA2 and SA3 are different from the corresponding data in Table 1.

10. Line 284-286, "and S1, S3, and S6 were influenced by both anthropogenic and wildfire CO emissions in North America (~56 %, ~79 %, ~40 %, and ~59 % for anthropogenic CO source, and ~42 %, ~19 %, ~53 %, and ~25 % for wildfire CO sources, respectively)."

    Check the values (there are four values for three samples).

11. Line 289. Change "Chemical-resolved" to "Chemically-resolved".

12. Line 292. Change ">400 particles cm-3" to ">400 particles cm$^{-3}$"

13. Line 296. "Our particles are internally mixed based on tilted transmission electron microscopy (TEM, the titled angle was 70°) (Fig. S8).", Line 328. "This observation is consistent with their STXM images and tilted TEM images (Fig. S8)".

    Give a more detailed explanation how an internal mixing state is determined?

    Line 297. Change "titled angle" to "tilted angle".

14. Line 298. "Fig. 2(b to i) show" should be Fig. 3(b to i).

15. Line 304. "area equivalence diameter" . Do you mean "area equivalent diameter"?

16. Line 306. The values of 79.6% and 1.1% did not match the values in Table S2.

17. Line 323. "Figure 4 shows the STXM-NEXAFS Carbon K-edge chemical speciation maps and spectra of four typical particle mixing states of OC (green), IN (blue), and EC (red) found in S3-3 and S4-2, which are (a) organic particle (green), (b) EC core (red) and coated by OC (green), (c) internally mixed EC (red) and In (cyan) coated by OC (green), and (d) In (cyan) coated by OC (green)".

    Do "blue" and "cyan" both indicate the inorganics? DO "IN" and "In" both indicate the inorganics? And the description here is different from the caption of Figure 4.

18. Line 331. "all samples"

    Check Figure S9 is for the results of all samples of only seven samples.

19. Line 332. "S3-3 and S4-4 samples"

   Do you actually mean S3-3 and S4-2 samples? I do not see S4-4 in Figure S9, and in Table S2, the sample analyzed by STXM-NEXAFS is sample S4-1. Also check the values that did not match the ones in Table S2.

20. Line 344. "Figure 5 shows violin plots of the 'corrected' aspect ratio (left) and representative tilted images (right) for Pico 2014 (a to c), Pico 2015 (d to i), and Pico 2017 (j to q)."

   The description here is different from the caption of Fig. 5. Correct it.

21. Line 368. "The substantial fraction of solid and semisolid particles might be less oxidized"

   In Table 1, I found that SA1 and S6, whose average aging time is both longer than 16 days, have smaller fraction of liquid particles than other samples. Can you explain why the fraction of liquid particles is smaller with longer aging time?

22. Line 379. Change "5(a, d, e, I, and j to o))" to "5(a, d, e, i, and j to o))"

23. Line 383. "For S4-2, a possible reason is that the volatile and less viscous species of particles collected on the TEM grid have already evaporated and left these tiny residuals around those big particles (see Fig. 5(f) right panel) due to difference in temperature, RH, and pressure between OMP and SEM chamber."
   Does this problem also exist in the experiments of other samples?

24. Line 402. I did not see the viscosity of BBOA predicted in DeRieux et al. (2018) is up to $10^{12}$ Pa s. I suggest you only show what is the range of the viscosity under the atmospherically relevant RH. Add Li et al. (2020) who also calculated the viscosity of BBOA based on volalilaty distributions (Li, Day et al. 2020).

25. Line 416. "Shaded areas represent regions of different phase states (liquid: blue, semisolid: green, and solid: red), with the boundaries of each region based on (O'Brien et al., 2014)."
   Can you give a more detaied explaination how to get the boundary lines?

26. Line 430. "We used the density ($\rho_{org}$), hygroscopicity ($\kappa_{org}$), and dry glass transition temperature ($T_{g,org,0}$) of organic particles as reported by Schum et al., 2018 (see SI) since we

do not have molecular compositions for our samples and Schum et al., 2018's samples were also collected at OMP during the same seasonal period (June and July).".

The previous analysis in this manuscript mentioned that the composition of organic matter is quite different for different samples. Therefore, $T_{g,org,0}$ would be changed. There are three samples in the study of Schum et al. (2018), and the estimated $T_g$ are also varied. Discussion of the uncertainties in $T_{g,org,0}$ is betted added.

27. Line 441, also cite (Schmedding, Rasool et al. 2020, Li, Carlton et al. 2021).

28. Line 490, cite (Li, Carlton et al. 2021, Shrivastava, Rasool et al. 2022).

29. Line 930. Change "solid black cycles" to "solid black circles"?

30. There is no need to use italics in the columns 12 and 13 in the first row in Table 1.

31. What does the colorbar in Figure 1 indicate?

32. The inserted figures should be described in the caption of Figure 3.

33. Change "SA1" to "SA2" for panel b in Figure 5.

**Supporting Information:**

34. Line 2. The title in the supplementary is different from the title in the manuscript.

35. Line 21. "where $T_{g,w}$ is equal to 136 K, is the $T_g$ for pure water".
    Cite (Kohl, Bachmann et al. 2005).

36. Line 29. "Moreover, $k_{GT}$, $T_{g,w}$, $\kappa_{org}$, and $\rho_{org}$ were assumed to be 2.5 (Shiraiwa et al. 2017), 309 K (Schum et al. 2018), 0.12 (Schum et al. 2018), and 1.4 g cm$^{-3}$ (Schum et al. 2018), respectively."
    Why 309 K is for $T_{g,w}$? Check it.

37. Figure S2. What the x-axis stands for in figures b to r?

38. In Figure S2 and Figure S3, are the relative atomic ratios of elements same as the relative element weight?

39. Figure S4. Change "Jun" to "June".

40. In Fig. S5-S6, I don't understand why the residence time is in percentage and how did you calculate it?

41. In the caption of Figure S10, "Mean ambient temperature (blue) and the predicted RH-dependent $T_{g,org}$ values (green)". The ambient $T$ is actually in green and $T_{g,org}$ is in blue in the figure.

**References**

Cheng, Y., H. Su, T. Koop, E. Mikhailov and U. Pöschl (2015). "Size dependence of phase transitions in aerosol nanoparticles." Nature Communications **6**: 5923.

Kohl, I., L. Bachmann, A. Hallbrucker, E. Mayer and T. Loerting (2005). "Liquid-like relaxation in hyperquenched water at [less-than-or-equal]140 K." Physical Chemistry Chemical Physics **7**(17): 3210-3220.

Li, Y., A. G. Carlton and M. Shiraiwa (2021). "Diurnal and Seasonal Variations in the Phase State of Secondary Organic Aerosol Material over the Contiguous US Simulated in CMAQ." ACS Earth and Space Chemistry.

Li, Y., D. A. Day, H. Stark, J. L. Jimenez and M. Shiraiwa (2020). "Predictions of the glass transition temperature and viscosity of organic aerosols from volatility distributions." Atmos. Chem. Phys. **20**(13): 8103-8122.

Li, Y. and M. Shiraiwa (2019). "Timescales of secondary organic aerosols to reach equilibrium at various temperatures and relative humidities." Atmos. Chem. Phys. **19**(9): 5959-5971.

Petters, M. and S. Kasparoglu (2020). "Predicting the influence of particle size on the glass transition temperature and viscosity of secondary organic material." Scientific Reports **10**(1): 15170.

Petters, S. S., S. M. Kreidenweis, A. P. Grieshop, P. J. Ziemann and M. D. Petters (2019). "Temperature- and humidity-dependent phase states of secondary organic aerosols." Geophys. Res. Lett. **46**.

Richards, D. S., K. L. Trobaugh, J. Hajek-Herrera, C. L. Price, C. S. Sheldon, J. F. Davies and R. D. Davis (2020). "Ion-molecule interactions enable unexpected phase transitions in organic-inorganic aerosol." Science Advances **6**(47): eabb5643.

Schmedding, R., Q. Z. Rasool, Y. Zhang, H. O. T. Pye, H. Zhang, Y. Chen, J. D. Surratt, F. D. Lopez-Hilfiker, J. A. Thornton, A. H. Goldstein and W. Vizuete (2020). "Predicting secondary organic aerosol phase state and viscosity and its effect on multiphase chemistry in a regional-scale air quality model." Atmos. Chem. Phys. **20**(13): 8201-8225.

Shrivastava, M., Q. Z. Rasool, B. Zhao, M. Octaviani, R. A. Zaveri, A. Zelenyuk, B. Gaudet, Y. Liu, J. E. Shilling, J. Schneider, C. Schulz, M. Zöger, S. T. Martin, J. Ye, A. Guenther, R. F. Souza, M. Wendisch and U. Pöschl (2022). "Tight Coupling of Surface and In-Plant Biochemistry and Convection Governs Key Fine Particulate Components over the Amazon Rainforest." ACS Earth and Space Chemistry **6**(2): 380-390.

---

## Author Comment (AC1)

We want to thank the reviewers for their comments. Addressing those comments has improved the quality of the manuscript. Below, we list each reviewer's comment (regular font), followed by our response (indented, **bold** font), followed by corresponding changes in the revised manuscript (indented, blue font). RL and RSL represent the line number in the revised main manuscript and SI, respectively.

**Anonymous Referee #1**

This manuscript by Cheng et al. collected samples over three years at an interesting site (North Atlantic). They also used various measurement techniques (e.g., CCSEM-EDS and STXM-NEXAFS) for a significant number of samples as well as modeling and provided a unique conclusion regarding particle phases. Thus, I think this study will be an interesting contribution to our understanding of atmospheric aerosol particles.

**We appreciate the positive feedback from the reviewer. Below are our responses to each comment:**

**Major comments.**

1. I suggest including a discussion regarding the effect of relative humidity (RH) on the particle phase. Aerosol particle phases are sensitive to the RH when collected (e.g., Bateman et al. 2014 in the reference list). Inorganic aerosol particles can deliquesce, and organic particles can absorb water depending on RH, changing the shapes of sampled particles. The RH values should be obtained from an in-site measurement, if available, (not from a model result with a low spatial resolution) as the particle hygroscopicity is sensitive to the exact RH during the sampling. Although most particles should be in dry condition judging from Table S2, hysteresis phenomena may affect the particle hygroscopicity (e.g., Fig. S10). The current manuscript has a limited discussion regarding the ambient RH, and I suggest more discussion on RH effects for the particle phases. In addition, surface tension may also influence the height of the aspect ratio of sampled particles, and some discussion regarding surface tension may be useful.

   > **We appreciate the reviewer for bringing up this point, which was not adequately discussed in the manuscript. We agree with the reviewer that particle phases are sensitive to relative humidity (RH) when collected. Based on the RH during sample collection, the shape of the particles will deform upon impaction, which we use to estimate their phase state (Cheng et al., 2021). We agree with the reviewer that the particles we investigated in this study might experience hysteresis phenomena that could affect the particle hygroscopicity and thus the phase state of particles during transport. RH-dependent phase state of ambient particles is an important topic and will be considered in future studies. However, as the reviewer suggested, we added the following sentences about RH-dependent phase states at the end of Line 375:**

   > RL406-411: "Besides these two potential explanations, many aspects can still affect the phase state of particles. Particles can transit from solid to semisolid to liquid state when RH and/or temperature increase (Koop et al., 2011). Thus, these particles might transit to different phase states if the ambient conditions change. For example, measured RH at OMP was highest during the S2 and S3 sample collection periods (61.3±2.4 % and 67.3±2.3 %, respectively) and lowest during the S4-2 and S4-3 collection periods (6.6±0.3 % and 9 %, respectively). The lower RH at OMP during S4-2 and S4-3 collection periods might help explain

the observation of more abundant solid state particles in S4-2 and S4-3 rather than S2 and S3."

**For the meteorological data, we added temperature and RH values measured at OMP during the specific sample collection periods (Table S2). However, meteorological data during transport were not experimentally accessible and were, therefore, extracted from the meteorology fields of the Global Forecast System (GFS) files (see Sect. 2.2) as the best option available to us. We agree that hysteresis phenomena can affect particle hygroscopicity, but these phenomena, at least during transport, cannot be investigated within the data availability of this study. As the reviewer suggested, we added the hourly variation of temperature and relative humidity as measured directly at OMP in the supplementary information (Fig. S2 in revision).**

**We also agree with the reviewer that surface tension plays an important role in the particle shape since different materials will have different surface tension, resulting in different contact angles at the substrate surface and liquid particle's surface. These are complex issues that depend on the properties of the particle material. However, we use Carbon Type-B TEM grids (Ted Pella Inc.) for our phase state assessment since our previous study (Cheng et al., 2021) was conducted with the same type of grids that have hydrophobic and oleophobic surfaces. Thus, the same aspect ratio threshold we found in our previous study should be appropriate. To make this point clear, we modified the sentence in Line 234:**

> RL251-254: "These thresholds were determined based on known RH-dependent glass transition of organic materials (e.g., Suwannee River fulvic acid (SRFA)) on the same grids type (Carbon Type-B TEM grids, Ted Pella Inc) used in this study (Cheng et al., 2021). Using the same grid type should minimize the effect of changes in surface tension and wettability, which might potentially affect the contact angle and therefore the aspect ratios."

2. The authors discuss the CO source contributions using the FLEXPART model. Although the model is acceptable and useful for CO, I wonder if it can be used to interpret the source of aerosol particles, especially for those with aging more than ten days. CO is gas and will not be removed from the atmosphere. On the other hand, a fraction of aerosol particles will be removed by mainly wet depositions during the transport with more than ten days (Table 1). Thus, it is not sure if the estimates of "contribution of source" in the table are valid for aerosol particles. Some explanation will be needed here.

**We agree with the review's comment on FLEXPART CO and the differences in chemical nature between CO and aerosols. We did not make it clear enough in the context that FLEXPART CO results in Table 1 are used as "*indicators*" of relative contributions from anthropogenic or biomass burning emissions and aging time during the transport instead of *quantitative estimates* of aerosol lifetime or mass. However, these indicators can reflect the aerosol sources and aging time because primary aerosols and aerosol precursors ($NO_x$, $NH_4$, BC, etc.) are heavily co-emitted with CO in anthropogenic and biomass burning emission sources. Comparisons of such indicators across the aerosol samples (in Table 1) reveal very useful information about the air mass source and transport history, which helps interpret the observed aerosol properties we got in the lab. Another reason for the long aging time (>10 days) reported for FLEXPART CO is due to the long simulation time we configured on**

**purpose. We track air mass transport back to 20 days for all samples. Aerosol lifetime against wet removal can be as short as a couple of days in the lower troposphere, but it can be extended to weeks in the free troposphere for long-range transport.**

**To make this clear, we have added the following:**

> RL263-265: "The plume ages and relative contributions from anthropogenic and biomass burning emissions can reveal air mass sources, types, and transport patterns. Although they do not directly reflect aerosol sources and ages, they are still good indicators to help interpret observed aerosol properties, especially in the comparisons across different samples."

3. Quality of Supporting information is a problem. The figures and captions include many errors, including the title (!), which is different from the manuscript. I wonder if the authors submitted the correct one or a draft version.

   **We do apologize for the quality of SI. We have revised the SI in the new submission.**

**Specific comments.**

4. Line 158. "an environmental SEM (ESEM) equipped with a FEI Quanta digital field emission gun, operated at 20 kV" and line 213 " Environmental Scanning Electron Microscopy (ESEM, Quanta 3D, Thermo Fisher)"

Are they different ESEM or the same one? The ESEM in line 158 is used for the CCSEMEDS? It isn't very clear, and please specify them clearly.

> **We used the same ESEM for both CCSEM-EDX and tilted imaging experiments. We revised line 158 and line 213 as below:**
>
> RL167-168: "We utilized an environmental SEM (ESEM, Quanta 3D, Thermo Fisher) equipped with a FEI Quanta digital field emission gun, operated at 20 kV and 480 pA."
>
> RL226-228: "We utilized tilted view imaging combined with the ESEM to estimate the phase state of particles based on their shapes. For each sample, we evaluated more than 150 randomly selected particles. Moreover, tilted view imaging and CCSEM-EDX experiments were performed independently."

5. Line 193 "inorganic components (In)"

In, IN, and "inorganics" are inconsistently used. For example, In is in line 207, "inorganics" is used in line 209, and IN is in line 324. In addition, "In" is confusing as it is like In (preposition).

> **We are sorry for the confusion. The revised manuscript uses IN for inorganic components retrieved from STXM-NEXAFS spectroscopy.**

6. Line 296-297. "Our particles are internally mixed based on tilted transmission electron microscopy (TEM, the titled angle was 70°) (Fig. S8)."

Please explain how to see Fig. S8, i.e., how the TEM image indicates internally mixed particles. Same for the description in line 328

**Thanks for bringing up this point. We have revised the manuscript:**

RL316-319: "Tilted transmission electron microscopy (tilted angle 70°) images show that inorganic inclusions (e.g., sea salt, nitrate, sulfate, dust) are internally mixed and coated by organics (Fig. S9)."

**And:**

RL351-352: "This observation is consistent with our tilted TEM images showing that EC and IN inclusions were internally mixed with organics (Fig. S9)."

**We also revised Fig. S8 (Fig. S9 in revision) and its caption:**

[Figure]

**Figure S9. Representative tilted transmission electron microscopy (TEM) images (tilt angle 70°) for S3-2. Green arrows indicate examples of thin organic coatings, and cyan arrows indicate examples of internally mixed inorganic inclusions (e.g., sea salt, nitrate, sulfate, dust, cycled by solid red lines) coated by organics.**

7. Line 317-319. "Sulfate (CNOS and sea salt with sulfate) particles are also very abundant in all samples (~18 to 34 %), suggesting that these particles were involved in cloud processing (Ervens et al., 2011; Kim et al., 2019; Lee et al., 2011, 2012; Zhou et al., 2019)."

I am not sure why they were involved in cloud processing. Sulfate can originate from various processes. Does it mean organosulfates (CNOS)??

**We appreciate the reviewer's valid comment, and we agree with the reviewer that sulfate can originate from various processes. With the CCSEM-EDX result, we cannot confirm if sulfates are organic. However, during the long-range transport of aerosol, the particles are expected to experience several cloud cycles. We modified the sentence as follows:**

RL338-340: "Sulfate (CNOS and sea salt with sulfate) particles are also abundant in all samples (~18 to 34 %), suggesting that these particles were possibly involved in cloud processing (Ervens et al., 2011; Kim et al., 2019; Lee et al., 2011, 2012; Zhou et al., 2019)."

8. Line 324-325. states of OC (green), IN (blue), and EC (red) found in S3-3 and S4-2, which are (a) organic particle (green), (b) EC core (red) and coated by OC (green), (c) internally mixed EC (red) and In (cyan) coated by OC (green), and (d) In (cyan) coated by OC (green).

Both "cyan" and "blue" are used for In. I think it should be blue or IN and In are different??

**Sorry for the confusion. We have corrected our content and used cyan for IN in Fig. 4**.

RL344-347: "Figure 4 shows STXM-NEXAFS Carbon K-edge chemical speciation maps and spectra for four typical particle mixing states of OC (green), IN (cyan), and EC (red) found in S3-3 and S4-2, which are (a) organic particle (green), (b) EC core (red) and coated by OC (green), (c) internally mixed EC (red) and IN (cyan) coated by OC (green), and (d) IN (cyan) coated by OC (green)."

9. Line 373-375. "These results suggest that apart from environmental factors, the inorganic components, the molecular weight of organic compounds, and the O/C ratio (or aging time) all affect the phase state of internally mixed particles."

They are true at specific RH values. For example, < RH 80%, ammonium sulfate is solid (crystal), and > RH 80%, they become liquid (deliquesce). These factors change the specific RH % that changes the particle phase state. Although it says "apart from environmental factors", some words about RH will be useful. Please see my comment 1.

**Please see our response to comment 1. We added a discussion about the ambient RH on the phase state of the particles.**

10. Line 409-410. "Typically, particles with the same area equivalent diameter but higher TCA are more viscous (more solid-like) since they are less flat in shape (Fraund et al., 2020; Tomlin et al., 2020)."

The particle height may be also influenced by its surface tension if they are liquid. Please see my comment 1.

**Please see our response to comment 1.**

11. Figure 1. Please indicate what are the color indicate and what are the boxes and numbers.

**We revised the caption of Fig. 1, Fig. S6, and Fig. S7 in revision:**

"Figure 1. Column-integrated residence time over the 20-day transport time retrieved from FLEXPART retroplumes for 2017. (a) S3-1, (b) S3-2, (c) S3-3, (d) S3-4, (e) S4-1, (f) S4-2, (g) S4-3, (h) S4-4. The vertical distribution of the retroplumes residence time at given upwind times are shown in Fig. S6. The color bars indicate the ratio of column integrated residence time to the maximal residence time at each upwind time in the logarithmic scales, and the X-axis and y-axis represent latitude and longitude, respectively. The numbers indicate locations of the highest vertically integrated residence time on a given upwind day."

"Figure S6. The vertical distribution of the retroplumes residence time at given upwind times retrieved from FLEXPART retroplumes for (a) S3-1, (b) S3-2, (c) S3-3, (d) S3-4, (e) S4-1, (f) S4-2, (g) S4-3, and (h) S4-4. The color bar represents the ratio of residence time to the highest residence time across the height scale at each upwind time. The black lines indicate the average height of the plumes during transport."

"Figure S7. Column-integrated residence time over the 20-day transport time and the vertical distribution of the retroplumes residence time at given upwind times retrieved from FLEXPART retroplumes for Pico 2015. (a, b) S1, (c, d) S2, (e, f) S3, (g, h) S4, (i, j) S5, (k, l) S6. For panels a, c, e, g, and i, the color bars indicate the ratio of column integrated residence time to the maximal residence time at each upwind time in the logarithmic scale, and the X-axis and y-axis represent latitude and longitude, respectively. For panels b, d, f, h, j, and l, the color bars represent the ratio of residence time to the highest residence time across the height scale at each upwind time, and the black lines indicate the average height of the plumes during transport."

12. Figure 2. These "solid black cycles" (circle?) are difficult to see with dark blue background.

**Based on all comments regarding Fig.2, we have revised Fig. 2 and its caption as below:**

[Figure]

**"Figure 2. (a) Normalized particle size distribution from 10 to 800 nm measured from SMPS measurements, and (b) SMPS derived total particle concentrations (left y-axis, red line) and CO tracer retrieved from FLEXPART simulations (right y-axis, blue line) from 05 to 21 July 2017."**

13. Figure 3. Although I can imagine what the inserted normalized number fractions with size distributions in the upper right of each panel mean, it is better to have some explanation, especially the meanings of Y-axes.

   **Thanks for pointing that out. We have revised the caption:**

   **"Figure 3. Chemically-resolved size distributions were inferred from the CCSEM-EDX data for 2017. (a) Fraction of different particle types for all samples. Normalized chemically-resolved size distributions of (b) S3-1, (c) S3-2, (d) S3-3, (e) S3-4, (f) S4-1, (g) S4-2, (h) S4-3, and (i) S4-4. Inserts represent the normalized number fraction of different particle types as a function of particle size."**

14. Figure 4. Please indicate which samples were used for each panel.

   **We revised the caption of Fig. 4:**

   **"Figure 4. Representative STXM-NEXAFS spectra of (a) organic particle (green), (b) EC core (red) coated by OC (green), (c) internally mixed elemental carbon (red), and IN (cyan) core coated by OC (green), and (d) IN (cyan) core coated by OC (green) from Pico 2017 S3-3 and S4-2 samples. White scale bars represent 500 nm."**

15. Figure 5. Is panel (b) SA1 or SA2?

   Table 1 indicates that 29.8% of SA1 particles are solid. Although I see SA1 includes relatively more semisolid particles, I cannot see solid particles. Could you indicate some examples of solid particles in the SEM images using ambient samples?

**We appreciated the reviewer's comment. We revised Fig. 5 to show the presence of solid particles. The revised portion is shown below:**

[Figure]

I also suggest adding RH values when collected for these samples.

**Thanks for your comments. We have corrected panel (b) from SA1 to SA2. We have included available RH and temperature in Table S1 and S2. We also added a plot that shows the available hourly variation of temperature and RH during the sampling days:**

[Figure]

**"Figure S2. Hourly variation of temperature and relative humidity for available days. Shaded areas represent the sample collection periods."**

16. Figure 6. In panel (a), there are 3 or 4 solid particles in SA2, but the solid particle % in SA2 is 0.0 in Table 1. Are they correct?

**Sorry for the confusion. In Table 1, the percentage of particles in each stage is determined by the titled image, which might have different results than the TCA estimated phase state due to technique differences and the difference in the investigated area on the grid. To make that clear, we modified the caption of Table 1 as below:**

"Table 1. Summary of Pico 2014, 2015, and 2017 samples. "S3-" and "S4-" for Pico 2017 samples were collected on stage 3 (50 % cut-off size: >0.15 μm) and stage 4 (50 % cut-off size: >0.05 μm) of a four-stage cascade impactor (MPS-4G1), respectively. Pico 2014 samples were collected on stage 3, and Pico 2015 samples were collected on stages 3 and 4 of the cascade impactor. Particle percentages of different phase state were retrieved based on tilted imaging. Additional information on sampling time and conditions and fractions of different species in each sample based on CCSEM-EDX and STXM-NEXAFS is listed in Tables S1 and S2."

**Supplementary information**

I do not think I could indicate all errors. Please check the data carefully (or maybe it is a wrong file?).

17. The title is different from the main text.

    **We updated the title and uploaded a new supplementary information file. We carefully reviewed the supplementary section and corrected all the mistakes we could find.**

18. Line 21. "where Tg,w is equal to 136 K, is the Tg for pure water," Tgw is 136K, correct? " is the Tg for pure water " is correct?

    **We have revised the sentence as below:**

    RSL51-52: "where $T_{g,w}$ is the $T_g$ for pure water, $k_{GT}$ is the Gordon-Taylor constant, $\kappa_{org}$ is the CCN-derived hygroscopicity parameter of the organic fraction, $\rho_{org}$ and $\rho_w$ are the density of water and organic material, respectively."

19. Equation S3. C_real=(123.2±1.4)−(4.738±0.214)log(H)−(1.186±0.02)C_measured.

    This equation indicates that less measured C atomic percentages yield a high "real" C percentage. I.e., if a particle includes no measured carbon percent (0%), it will have ~100 % of real C percent (by assuming H = 1). Although I do not have a way to check the accuracy, it is difficult to believe the result without more explanation. The calculation may influence the results in Figure S2, in which a fraction of particles consists of only C (no O nor other elements).

    The equation S4 is also questionable. How can O=0%, which is seen in Fig. S2, be achieved?

    O_real=(13.68±0.18)−(0.3413±0.0636)log(H)+(0.2579 ± 0.0072)O_measured (S4)

    **Thanks for pointing this issue out. Theoretically, we can get 100% of C even if we do not detect any C by assuming H = 1. However, if we do not detect C and O, or C and**

**O are equal to 100%, we do not do any correction. Moreover, if corrected C and O fractions are either smaller than 0 or larger than 100%, we will discard these data since they are not realistic. We admit this correction method has limitations, and it is based on statistical analysis with some assumptions that CNQX disodium salt particles are perfect spheric, and all particles have the same aspect ratio. However, this is still a reasonable method and provides a better estimate than the raw data. We agree with the reviewer that some explanations would help to clarify the limitations and assumptions. To this aim, we added the following sentence in SI:**

> RSL36-44: "Moreover, we only perform this correction when $C_{measured}$, $O_{measured}$, and $N_{measured}$ are not equal to 0 or 100% since these cases are not realistic. Furthermore, if corrected C, N, and O values are less than 0 or greater than 100%, we discard these data since they are also not realistic. Therefore, we applied this correction to measured C, N, and O, and after correction, we re-normalized the fraction of all elements. It should be kept in mind that this correction method is based on empirical fittings with assumptions that CNQX disodium salt particles are perfect spheric, and all particles have the same aspect ratio. The first assumption might lead to overestimating the particle height of CNQX disodium salt particles, and the second one might misrepresent the particle shape. Moreover, using one standard might not fully represent the chemical complexity of ambient particles. Thus, more data from different standards are necessary for improving this method."

20. Line 49-51. "Since the particles are spheric, the measured area equivalent diameter (μm) is approximately equal to the height of particles. Therefore, when applying the correction function on our CCSEM-EDX data, we need to estimate the H by dividing the longest diameter retrieved from CCSEM-EDX measurement by the aspect ratio retrieved from tilted images (see Sect. 3.3.2). "

Do you have all aspect ratio data for all EDS measured particles? I think the aspect ratio was measured using ESEM, and the EDS was by CCSEM-EDS.

> **Thanks for your comment. We have all aspect ratio data from CCSEM-EDX measurements from the top-view measurement. We used the same instrument (ESEM) for tilted view imaging and CCSEM measurements. However, these measurements are performed separately. The current configuration of the instrument does not allow for simultaneous measurements. We added the following text to the revised manuscript.**
>
> > RL227-228: "Moreover, tilted view imaging and CCSEM-EDX experiments were performed independently."

21. Table S1. Are there CCSEM data that can be listed for these samples?

> **Thanks for your comment. CCSEM data are reported in table S2 for 2017 samples, and data for 2014 samples are already published (Lata et al., 2021). Data for 2015 samples will be published in a separate manuscript focusing on aerosol optical properties. All CCSEM-EDX raw data are available upon request.**

22. Figure S2. These data, especially for C, look different between those from SA1 to S6 and those from S3-1 to S4-4 (different sampling periods). Are there any technical differences?

Potassium (K) may be used for a biomass-burning tracer. Have you checked it?

**We appreciate that reviewer brought this question up. We agree with the reviewer that the C looks different for SA1 to S6 (collected in 2014 and 2015) than S3-1 to S4-4 (collected in 2017). Data were acquired with the same instrument and same configuration (e.g., same working distance, accelerating voltage, and beam current), so we think that there is no technical difference rather than the difference in the sample itself. We also agree with the reviewer that potassium is a good indicator of biomass burning emission. However, the elemental percentage of K is very low (less than 0.5%), below the sensitivity of the measurements. This is why we did not specifically use K as a tracer for this study.**

23. Figure S3. If you go to "No" and "No," you will find a question "Al+Si+Fe+Fe>Na", where you have double Fe.

**Thanks for pointing this out. We have corrected Fig. S3 (Fig. 4 in revision) to represent the right particle classification logic.**

24. Figure S4. Panel (a). There is "S-2," but it should be "S3-2." Y-axis should have "100" instead of "00". The caption should be "June" instead of "Jun."

**Thanks for your comment. We have revised Fig. S4 (Fig. S5 in revision) and its caption.**

[Figure]

**"Figure S5. FLEXPART CO tracer simulation for (a) June 2017 and (b) July 2017."**

25. Figure S5. The caption indicates from (a) to (i), whereas the panels are from (a) to (h).

    **Thanks for your comment. We revised the caption as below (Fig. S6 in revision):**

    **"Figure S6. The vertical distribution of the retroplumes residence time at given upwind times retrieved from FLEXPART retroplumes for (a) S3-1, (b) S3-2, (c) S3-3, (d) S3-4, (e) S4-1, (f) S4-2, (g) S4-3, and (h) S4-4 for Pico 2017. The color bar represents the ratio of residence time to the highest residence time across the height scale at each upwind time. The black lines indicate the average height of the plumes during transport."**

26. Figure S7. "Jun" should be "June." Panel (a) and (b) is upside down. The legend in the panel (a, bottom) is overlapped with the plot.

    **Thanks for pointing this out. We have revised Fig. S7 (Fig. S8 in revision) and its caption as below:**

[Figure]

"**Figure S8. FLEXPART CO tracer simulation for (a) June 2015 and (b) July 2015.**"

27. Figure S8. Please indicate where we should see. Please see my comment 6.

   **Thanks for your valid comment. We added cyan arrows in Fig. S9 to indicate inorganic inclusions and green arrows to indicate organic coatings. We have revised the caption of Fig S8 (Fig. S9 in revision).**

   "**Figure S9. Representative tilted transmission electron microscopy (TEM) images (tilt angle 70°) for S3-2. Green arrows indicate examples of thin organic coatings, and cyan arrows indicate examples of internally mixed inorganic inclusions (e.g., sea salt, nitrate, sulfate, dust, cycled by solid red lines) coated by organics.**"

28. Figure S9. The colors in OCInEC and In are nearly the same and cannot be distinguished. For example, in panel (f), it is difficult to identify if the light blue is OCInEC or In.

**Thanks for your comment. We have revised Fig. S9 (Fig. S10 in revision) as below:**

[Figure]

29. Figure S10. "Mean ambient temperature (blue)"

In the caption, the temperature is "blue," but in the legend, it is green. Same for Tg,org.

"(g) S3-2, (g) S4-3, (h) S4-4, and (i) S4-54. " There are two (g) in the caption. (i) should be S4-5 but no (i) in the panel (!!). "uncertainties in RH (See SI). " Which SI should we see. we are now in SI.

**Thanks for your comment. We have revised the caption of Figure S10 (Fig. 11 in revision):**

**"Figure S11. Mean ambient temperature (green) and relative humidity (RH) (red) extracted from the GFS analysis along the FLEXPART modeled path weighted by the residence time and the predicted RH-dependent $T_{g,org}$ values (blue) for (a) S3-1, (b) S3-2, (c) S3-3, (d) S3-4, (e) S4-1, (f) S3-2, (g) S4-3, and (h) S4-4. The blue and red shaded areas represent one standard deviation of ambient temperature and RH from the GFS analysis. The green shaded areas represent uncertainties of predicted $T_{g,org}$ estimated from the range of $T_{g,org}(RH = 0\%)$ and uncertainties in RH."**

30. References. The reference style is different from that of ACP.

**Thanks for pointing out this. We have corrected the reference style.**

31. Line 134 "Zieger, P. and Va, O" Please check the authors' name.

**Thanks for mentioning this, and we have removed this reference since we do not cite it in the text.**

---

## Author Comment (AC2)

We want to thank the reviewers for their comments. Addressing those comments has improved the quality of the manuscript. Below, we list each reviewer's comment (regular font), followed by our response (indented, **bold** font), followed by corresponding changes in the revised manuscript (indented, blue font). RL represents the line number in the revised version.

**Anonymous Referee #2**

The authors described the phase states of aerosol particles collected in the North Atlantic FT and tried to explore the transport patterns of the aerosol particles. Such research topic is interesting for the atmospheric communities, and also the scope of the research is suitable in ACP journal. However, after carefully reviewing this manuscript, the evidence are rather weak to support the results, and conclusion is too generalized. In addition, many errors in the text, figures, Tables, and SI can be founded.

> **We appreciate the constructive feedback from the reviewer. We attempted to streamline some conclusions and noted the study's limitations. Below are our responses to each comment:**

**Major comments:**

1. During the laboratory experiments for the phase determination, at which relative humidity and temperature the ESEM did the authors perform? This should be clearly stated in the manuscript. The main issue is that how the authors can conclude the phase states of the aerosol particles if the relative humidity and temperature during the experiments were different compared to the field measurement periods? The phase states of aerosols are temperature- and relative humidity-dependent, and thus it didn't convince me whether the conclusion is still valid or not. This should be clearly mentioned through the manuscript. The authors should also show the ambient RH and temperature at the monitoring site in a figure and table.

> > **We appreciate that reviewer pointed this out. Section 2.3 RL167 mentioned that "*Ambient particle samples were analyzed with ESEM at 293 K and under vacuum conditions (~2×10$^{-6}$ Torr)*". These conditions are not representative of the ambient atmosphere. Our measurements capture the phase state of particles at the time of sample collection. We agree that, in principle, changes in temperature and humidity in the ESEM chamber (under vacuum conditions, the RH inside the chamber is close to 0%) could affect the phase state of an airborne particle. However, our inference of the particle's phase state at the time of collection is based on the shape the particle acquires at impaction on the substrate, which unlikely would change significantly within the ESEM chamber due to adhesion forces between the particle and the substrate. This is a caveat of this method, which has been reported in previous studies. (e.g., Cheng et al., 2021; Lata et al., 2021; Wang et al., 2016); however, we believe that these results still provide useful information about the phase state of the particles in the atmosphere. Future studies should focus on determining the uncertainties introduced by RH-dependent phase states. To make this point clear, we add the following sentences in L167:**

> > > RL167-174: "Ambient particle samples were analyzed with ESEM at 293 K, under vacuum conditions (~2×10-6 Torr) and therefore at RH values near zero, which might lead to losses of volatile and semivolatile materials. Moreover, the temperature and RH inside the ESEM chamber differed from those at the OMP during sample collections (Fig. S2). RH and T affect the phase state of airborne particles; however, our inference of the particle's phase state at the time of

collection is based on the shape the particle acquires at impaction on the substrate, which unlikely would change significantly within the ESEM chamber due to adhesion forces between the particle and the substrate. These limitations need to be considered when interpreting our results."

[Figure]

**Figure S2. Hourly variation of temperature and relative humidity for available days. Shaded areas represent the sample collection periods.**

2. Regarding the technique of the tilted aspect ratios to determine the phase state of aerosols, I am confusing this technique is reliable for aerosols consisting of mixtures of organic materials and inorganic compounds. The authors should validate and carefully described the evaluation of the results with comparison to previous phase studies using well-known mixtures or commercial standards comprising organic and inorganics. I cannot find such validation from Cheng et al. 2021.

**We appreciate that reviewer for pointing this out. The technique of tilted aspect ratio has been used to study ambient particles in previous studies (e.g., Cheng et al., 2021; Fraund et al., 2020; Lata et al., 2021; Sharma et al., 2018; Tomlin et al., 2020; Veghte et al., 2017; Wang et al., 2016). We agree with the reviewer that the aspect ratio thresholds used to define the phase state were based on standard organic material, but we showed that the technique was also applicable to field-collected samples in Cheng et al., 2021. In addition, we also utilize the STXM/NEXAFS measurements to determine the phase state, which was also applied in previous studies (O'Brien et al., 2014; Tomlin et al., 2020). We agree with the reviewer that mixtures of organic and inorganic standards would be useful to refine those thresholds further. However, we think these results are still valuable to assess the phase state of individual submicron size particles, and future studies should focus on measurements of phase state from tilted view imaging. Thus, we plan to directly measure the viscosity from standard organic-inorganic mixtures in a future study.**

**To clarify the limitation of the study, we added the following sentences in the revised manuscript:**

> RL549-555: "This study assesses the phase state of internally mixed FT particles at the time of sample collection, and highlights the importance of accounting for inorganic inclusions to evaluate the phase state of internally mixed particles. Our results might not fully represent the phase state of FT particles during transport due to differences in ambient temperature and RH. Moreover, the aspect ratio thresholds used to determine the particles' phase states are based on limited standards. Future studies should focus on improving the aspect ratio thresholds by using more standards with known viscosities and determining the viscosity of internally mixed individual particles as a function of temperature and RH."

3. Figures and SI should be revised (see also below). Moreover, all figures in SI should be mentioned in the main text.

> **We thank the reviewer for this comment. We have made corrections in the revised version.**

**Minor comments:**

4. Page 5 line 136: The author should provide more details about stored conditions by mentioning temperature. Furthermore, the authors have to mention the stored period before the experiment due to evaporation issue.

> **In the main manuscript RL138-139, we mentioned that "Samples were stored at ambient condition and wrapped in Al foil immediately and kept in zip lock bags after collection to avoid exposure to light and air and minimize potential modification and oxidation in the air." Moreover, Pico 2017 samples were analyzed in the same year as soon as we received them from OMP. We agree that some sample modifications might occur during storage, but this is a limitation for any offline analysis of field-collected samples. We also underline that the site is quite challenging to access, making prompt sample analysis much more challenging than other sites; however, this aspect also makes the samples particularly unique and valuable. Thus, we add the following sentence in L137 in the revised manuscript:**

> RL141-145: "Samples were placed in dedicated storage boxes wrapped in Al foil and kept in zip lock bags immediately after collection to avoid exposure to light and outside air. The samples were then stored at ambient conditions to reduce the chances of modifications and oxidation that might have partially intercurred. This is a limitation of any offline analysis of field samples. We underline that the site is quite difficult to access; therefore, samples were delivered and analyzed as soon as it was feasible (less than one year after collection)."

5. The authors should provide details about the particle regeneration in the experimental section if it regenerated from the collected samples.

> **We did not regenerate any particles in this study. Particles were collected on TEM grids (carbon-B film), and analyses were performed directly on those grids.**

6. There are too many academic terms in the manuscript and it is suggested to add a table to summarize all acronyms and full names. The authors repeatedly used a similar abbreviation for the OC component with different names such as Organic (OC) (Page 7 line 192), and organic carbonaceous (OC) (Page 6 line 170). Abbreviation similarity should be consistent without repetition.

> **We thank the reviewer for this great suggestion. We have added a Table of Acronyms and Abbreviations (Table 2) in the main manuscript and revised L170 as below:**
>
>> RL183-185: "Based on their element percentage, each particle in Pico 2017 can be classified as organic (OC), carbonaceous with nitrogen (CNO), carbonaceous with sulfate (CNOS), sea salt (Na-rich), sea salt with sulfate (Na-rich with S), dust (Al, Si, Ca, Fe), dust with sulfate (Al, Si, Ca, Fe, S), and others (see Fig. S4)."

7. Page 4 line 119: The author mentioned "This study focuses on detailed individual particle analysis on Pico 2017". In addition, on page 6 line 172, the authors mentioned "CCSEM-EDX based particle classification for Pico 2014 can be found in Lata et al., 2021, and that for Pico 2015 will be discussed in our future work". However, some data relevant to the phase state for the 2014 and 2015 shown in Fig. 5. Also, total carbon absorption (TCA) data showed in fig 6 for Pico 2014, and Pico 2015. It makes confusion to the readers regarding which data Pico 2014, Pico 2015, or Pico 2017 is exactly discussed in this manuscript. To avoid more confusion author has to focus more on Pico 2017 data or the data relevant to Pico 2014 and Pico 2015 should move to SI.

> **We agree with the reviewer that all the data might be confusing for the readers without proper description. We tried to address this comment in our revised manuscript. The CCSEM-EDX analysis and FLEXPART simulation for Pico 2014 has been discussed in detail in Lata et al., 2021, and these for Pico 2015 will be discussed in future work. The main focus of this manuscript is the phase state of particles in the North Atlantic free troposphere during summertime based on samples collected over three different years. Thus, the tilted view imaging (Fig. 5) from ESEM and TCA analysis from STXM (Fig. 6) are crucial. However, to help our discussion of the association between chemical composition and source contributions with phase state, we decided to add short summaries of tilted view imaging analysis and TCA analysis from Pico 2014 and 2015. Therefore, we believe keeping Fig. 5 and Fig. 6 with data from all three samples can help generalize the findings and help the reader understand our main findings. We modified the text as follows in the revised manuscript RL116-122.**
>
>> RL117-123: "In this study, we present an overview of the phase state of individual FT atmospheric aerosol particles collected at OMP over three different years, which are July 2014 (Pico 2014), June and July 2015 (Pico 2015), and 2017 (Pico 2017). Analysis of samples from three years using tilted view Environmental Scanning Electron Microscope imaging and scanning transmission X-ray microscopy with near-edge X-ray absorption fine structure spectroscopy (STXM-NEXAFS) are reported to study the phase state of individual particles. The chemical composition and phase state of individual particles for Pico 2014 have been reported in a previous study (Lata et al., 2021). The chemical composition of individual particles for Pico 2015 will be discussed

in future work. This study focuses on detailed individual particle analysis of the Pico 2017 samples."

8. Page 8: In the result and discussion section, the description of Fig. 1 looks confusing and keeps the reader browsing to keep up with the text. The text is littered with redundant statements in parentheses that re-state what has just been explained. Please specify them clearly.

**We appreciate this comment from the reviewer and are sorry for the confusion. This study discussed eight FLAXPART retroplume analyses and the related CO tracer simulation results for the period the Pico 2017 samples were collected over, which were not discussed before. Some of the statements in parentheses have indeed been provided in Table 1, but we believe showing the numbers in the main body of the paper can help readers understand the contribution of CO from different sources, which are important for our discussion in Section 3.3.1. Nevertheless, we agree with the reviewer that it might be difficult for readers to keep up with the text. We attempted to reduce the redundant statements.**

9. Page 10 lines 290-294: More careful and detailed description are needed for Fig. 2 by comparing it with the reported study because size distribution is a very important factor when defining the physicochemical properties of an ambient particle. Also, please add how you measured in Experimental.

**Thanks for the comment. We agree with the reviewer that size distribution is a critical factor in understanding ambient particles' physicochemical properties. The detail of SMPS measurements has been described in (Siebert et al., 2021), which reports details on the Azores Stratocumulus Measurements of Radiation, Turbulence and Aerosols (ACORES) campaign in July 2017 that took place in the Azores, including activities at OMP. Therefore, based on the reviewer's suggestion, we revised L290-294 as below:**

RL303-313: "Figure 2 shows the particle size distribution and the total particle concentration based on SMPS measurements at OMP, and CO tracer concentrations in the air masses that arrived at OMP as retrieved from FLAXPART simulations (5 July 2017 to 21 July 2017). Mobility diameter ranged from 30 nm to 500 nm, and the mode was around 60±22 nm (Fig. 2a). The total particle concentration was around 279±114 # cm$^{-3}$. The size range, size mode, and particle concentration were comparable to those found in previous studies for FT particles (10-1000 nm, <100 nm, $10^1$ to $10^4$ # cm$^{-3}$, respectively) (Igel et al., 2017; Rose et al., 2017; Sanchez et al., 2018; Schmeissner et al., 2011; Sun et al., 2021; Venzac et al., 2009; Zhao et al., 2020). Figure 2b shows that the total particle concentrations positively correlate with the CO tracer concentrations from July 5th to July 12th and from July 18th to 21st, suggesting the major sources of particles during these periods might be anthropogenic and wildfire emissions. On the other hand, particle concentrations between late July 12th and 17th were above 279 # cm$^{-3}$, while the CO tracer level was relatively low (<10 ppbv) compared to other days, which might indicate additional sources of particles (e.g., sea spray and dust)."

**We also added more details regarding the SMPS measurements in the Experiment section:**

RL145-147: "Moreover, from 05 July to 21 July 2017, we also deployed a Scanning Mobility Particle Sizer (SMPS, TROPOS, for details, see Wiedensohler et al., 2012 ) coupled with a silica gel diffusion dryer to monitor the dry particle size distribution (<40% RH) and the total particle concentration with 5 mins time resolution (Siebert et al., 2021)."

10. To make this manuscript understandable to the readers, I would like to suggest the authors move data relevant Pico 2014, Pico 2015 to the supporting information. It has been already published.

    **We thank the reviewer for this valid suggestion. Please see our response to comment 7.**

11. The authors didn't describe clearly which samples were used for Fig. 4 which is relevant to STXM/NEXAFS spectra, Is that data relevant to Pico 2017? Even though there is no clear evidence in the description part (Page 11 line 323 to 329).

    **Thanks to the reviewer for this suggestion. These particles were from S3-3 and S4-2 from Pico 2017. These are representative particles to help readers understand typical STXM-NEXAFS spectra and maps for four types of particles. We have revised the manuscript:**

    RL344-347: "Figure 4 shows STXM-NEXAFS Carbon K-edge chemical speciation maps and spectra for four typical particle mixing states of OC (green), IN (cyan), and EC (red) found in S3-3 and S4-2, which are (a) organic particle (green), (b) EC core (red) and coated by OC (green), (c) internally mixed EC (red) and IN (cyan) coated by OC (green), and (d) IN (cyan) coated by OC (green)."

    **And:**

    **Figure 4. Representative STXM-NEXAFS spectra of (a) organic particle (green), (b) EC core (red) coated by OC (green), (c) internally mixed elemental carbon (red), and IN (cyan) core coated by OC (green), and (d) IN (cyan) core coated by OC (green) from Pico 2017 S3-3 and S4-2 samples. White scale bars represent 500 nm.**

12. Please clarify the captions of the SI.

    **We appreciate the reviewer for pointing this out. We have made the necessary changes to make the captions of the SI clearer. Please check the revised version.**

13. The title should be revised based on the main findings.

    **We updated the title in the SI, and it is now the same as the main text. Our title is based on the main findings.**

---

## Author Comment (AC4)

We want to thank the reviewers for their comments. Addressing those comments has improved the quality of the manuscript. Below, we list each reviewer's comment (regular font), followed by our response (indented, **bold** font), followed by corresponding changes in the revised manuscript (indented, blue font). RL and RSL represent the line number in the revised main manuscript and SI, respectively.

**Anonymous Referee #3**

Cheng et al. made a comprehensive investigation on the sources, chemical composition, and phase state of long-range transported free tropospheric particles based on individual particle analysis using multi-modal micro-spectroscopy techniques. This study found that most particles were in the liquid state and highlighted the importance of considering the mixing state, emission source, and transport patterns of particles when estimating particle phase state in the free troposphere. Though the findings are expected, the observation data provide valuable information constraining the physiochemical properties of aerosols in the free troposphere which is important in assessing aerosol associated climate effects. I agree with the comments of the other two referees that this manuscript can be published only after a major revision as there are too many errors in the submitted version.

> **We appreciate the positive feedback and constructive criticism from the reviewer. We attempted to fix all the errors in the manuscript. The reviewer raised some critical points, which we believe were addressed in the revised version and strengthened the article. Below are our responses to each comment:**

**Major comments:**

1. **We want to thank the reviewer for these important comments. Please see our response to each of points below:**

   a. My major concern is on the Section 3.3.2, the phase state of particles during long-range transport. The authors mainly investigated the phase state of organic particles applying the temperature and RH along the air mass transport path, and found that organic particles would likely be solid in most of the times. As the particle viscosity depends significantly on RH as pointed by the other two referees, I doubt the meaningfulness investigating the phase state at each air mass path with a wide variation in RH as shown in Fig. S10.

   > **We agree with the reviewer that temperature, relative humidity, and chemical composition of particles in the free troposphere are highly variable, and the phase state and viscosity of those particles depend on these factors, leading to large uncertainties. It would be ideal to measure the ambient conditions of air masses and phase state and viscosity of FT particles at multiple points over their transport path, but that is not feasible. Therefore, we resort to models to predict RH and T during transport. As shown in Fig. S6 in the revised paper, the vertical dispersion of air masses increases significantly after more than 5 upwind days, leading to significant variations in retrieved temperature and RH values. To minimize the induced uncertainties in the glass temperature calculations, we only predicted $T_{g,org}$ up to 5 days before the air mass reached OMP. We acknowledge these limitations in the current version of the manuscript; however, we still believe that the analysis provides valuable information to understand the evolution of the phase state of FT organic particles. We note that a similar approach was also adopted in a previous publication (i.e., Schum et al., 2018 ACP). To make this clear, we add the following sentences in RL481-483:**

RL484-486: "The estimates only up to 5 days before air masses arrived at OMP to avoid increased uncertainties associated with possible meteorological conditions due to the spread of the air masses (see Fig. S6)."

b.  I suggest adding a figure showing the variation of predicted viscosity with RH and T (similar to the figures in (Li and Shiraiwa 2019, Petters, Kreidenweis et al. 2019)) and investigating the phase state at free troposphere-relevant conditions.

**As the reviewer suggested, we added Fig. S12 in SI and a sentence in RL484-485.**

[Figure]

RL486-487: "Predicted $T_{g,org}$ to $T$ ratio as a function of temperature and relative humidity are shown in Fig. S12."

**"Figure S12. $T_{g,org}/T$ ratio as a function of temperature and relative humidity for organic particles transport in FT by using (a) minimum,**

**(b) median, and (c) maximum dry glass transition temperatures of organics as reported in Schum et al., 2018 (313.5 K, 332.1 K, and 360.65 K, respectively)."**

c. In addition, the authors applied a single value for the dry glass transition temperature, which, however, would be changed due to the change in the chemical composition during the long range transport.

**As mentioned earlier, we fully agree with the reviewer that dry glass transition temperatures will differ for different chemical compositions, which might change during the long-range transport. To study the sensitivity of the dry glass transition temperatures to different chemical compositions, we used the minimum and maximum dry glass transition temperatures of organic particles as reported in Schum et al., 2018 (360.65K and 313.46K, respectively). The ranges of glass transition temperatures are shown as the shaded areas of predict $T_{g,org}$ in Fig. S11 in the revision, resulting from variations in temperature, RH, and chemical composition.**

d. Finally, could the authors add some discussion that based on the inorganic component types you have observed, how you expect the phase state variation of inorganic components during the long range transport at free tropospheric RH and T? It would be helpful supporting the implication what you wrote in the Conclusion section that the particles in the FT probably remain liquid.

**We thank the reviewer for pointing this out. In Sect. 3.2, we have shown that our samples were internally mixed with hygroscopic salts such as sea salt and sulfate. Therefore, we expected that the presence of these inorganic inclusions would reduce the viscosity of FT particles at the RH and T values encountered in the FT during transport. This has been discussed in the last paragraph in Sect. 3.3.2. To make this clearer to the readers, we added the following sentence in the revised manuscript:**

RL509-512: "Thus, our results suggest that estimating the phase state of particles without considering the mixing state of FT particles might not accurately predict their viscosity and $T_g$ because the presence of hygroscopic inorganic inclusions (e.g., sea salt and sulfate) can reduce the viscosity of FT particles at the RH and T values encountered in the FT during transport."

**We further added one sentence in the conclusion section:**

RL541-542: "Moreover, the fraction and chemical composition of inorganic inclusions may further influence the phase state variations."

2. Mixing state plays an important role in the phase state of ambient particles; however, the authors did not mention other factors that may impact phase state significantly. Besides the influences of surface tension on aspect ratio and thus the prediction of phase state mentioned by Referee #1, the influence of particle size should be considered and discussed as well. Several studies have found that the size of particles influence the viscosity (Cheng, Su et al. 2015, Petters and Kasparoglu

2020). Did you see the difference in the phase state between the particles collected on the 3rd and 4th states of the impactor? Would the change of particle size affect the phase state during the long range transport? Secondly, the authors only mentioned the inorganic components could decrease the viscosity of internally mixed particles. They missed a recent study showing that increasing inorganic fraction can increase aerosol viscosity through cooperative ion-molecule interactions (Richards, Trobaugh et al. 2020).

**We want first to thank the reviewer for these very constructive comments. Please see our responses to your questions below:**

a.  Did you see the difference in the phase state between the particles collected on the 3rd and 4th states of the impactor? Would the change of particle size affect the phase state during the long range transport?

> **We did look at the size dependence of viscosity. In this study, SA1-SA3 and S3-1-S3-4 were collected on stage 3, and S4-1 to S4-4 were collected on stage 4. Since these samples were collected during different times, we are reluctant to draw any firm conclusions about the dependence of viscosity and phase state on particle size differences based only on these samples. On the other hand, samples S1-S6 were collected on stage 3 and stage 4 simultaneously, and we revised Fig. 5 (the figure below shows the revised portion of Fig. 5) to show the stage difference of aspect ratio distribution for these samples. As shown in revised Fig. 5, aspect ratios are different between stage 3 and stage 4. The particles collected on stage 3 for samples S1, S2, S4, and S6 have a higher fraction of more viscous particles than those collected on stage 4. However, particles collected on stage 3 for samples S3 and S5 have lower fractions of more viscose particles than those collected on stage 4. Therefore, we do not see a clear trend in the size dependence of the particle viscosity. However, we agree with the reviewer that this is worth further study. Hence, we added the following discussion in the revised manuscript:**

> > RL413-421: "Moreover, Cheng et al., 2015, Petters and Kasparoglu, 2020, and Kaluarachchi et al., 2022 have shown that particle size also affects the particles' viscosity. This appears to be the case for some samples when comparing the aspect ratio distribution for the Pico 2015 particles collected on stage 3 (left violin plots, 50 % cut-off size is >0.15 μm) with those from stage 4 (right violin plots, 50 % cut-off size is >0.05 μm) in Fig. 5d to 5i. For samples S1, S2, S4, and S6, particles from stage 3 have lower mode and mean aspect ratio than those from

stage 4, indicating that larger particles have higher fractions of more viscous particles than smaller particles. However, the aspect ratio distributions for particles collected on stage 3 in samples S3 and S5 have higher modes and mean values than those on stage 4, suggesting a higher fraction of less viscous particles in S3 and S5 on stage 3. Due to these inconsistent observations and the limited number of samples, we cannot draw clear conclusions regarding the size dependence of particle viscosity; this important aspect should be the objective of future studies."

[Figure]

b. Secondly, the authors only mentioned the inorganic components could decrease the viscosity of internally mixed particles. They missed a recent study showing that increasing inorganic fraction can increase aerosol viscosity through cooperative ion-molecule interactions (Richards, Trobaugh et al. 2020).

**Thanks for pointing out this paper. Richards et al., 2020 show that divalent ions (e.g., $Mg^{2+}$ and $Ca^{2+}$) can enhance the viscosities of organics through ion-molecule interactions. Unfortunately, we cannot identify the chemical formula involving these divalent ions with our technique. However, in our samples, the total percentage of these elements is very low (~0.17±0.34 %), while Na (~0.61±1.0 %) and sulfate**

(~0.48±0.40 %) are more abundant. Thus, the effect of ion-molecule interactions might not be critical for the particles investigated in our study. However, we agree that this is an important mechanism that should be discussed in the manuscript. To make this clear, we revised the manuscript and added the discussion below:

> RL379-382: "Besides, Richards et al., 2020 have reported that divalent ions (e.g., $Mg^{2+}$ and $Ca^{2+}$) can increase aerosol viscosity due to ion-molecule interactions. Although our analytical technique cannot identify the chemical formula involving these divalent ions, this phenomenon might not be critical for our samples because we found only minor fractions of Mg and Ca."

**Specific comments:**

**Manuscript:**

1. I recall the comments by the other two referees that the RH in the ESEM should be clearly pointed out as the particle phase state depends significantly on RH.

   **We thank the reviewer for making these valid comments. Please refer to our response to reviewer 2, major comment 1:**

   **We appreciate that reviewer pointed this out. Section 2.3 RL167 mentioned that "*Ambient particle samples were analyzed with ESEM at 293 K and under vacuum conditions (~2×10⁻⁶ Torr)*". These conditions are not representative of the ambient atmosphere. Our measurements capture the phase state of particles at the time of sample collection. We agree that, in principle, changes in temperature and humidity in the ESEM chamber (under vacuum conditions, the RH inside the chamber is close to 0%) could affect the phase state of an airborne particle. However, our inference of the particle's phase state at the time of collection is based on the shape the particle acquires at impaction on the substrate, which unlikely would change significantly within the ESEM chamber due to adhesion forces between the particle and the substrate. This is a caveat of this method, which has been reported in previous studies. (e.g., Cheng et al., 2021; Lata et al., 2021; Wang et al., 2016); however, we believe that these results still provide useful information about the phase state of the particles in the atmosphere. Future studies should focus on determining the uncertainties introduced by RH-dependent phase states. To make this point clear, we add the following sentence in L161:**

   > RL167-174: "Ambient particle samples were analyzed with ESEM at 293 K, under vacuum conditions (~2×10-6 Torr) and therefore at RH values near zero, which might lead to losses of volatile and semivolatile materials. Moreover, the temperature and RH inside the ESEM chamber differed from those at the OMP during sample collections (Fig. S2). RH and T affect the phase state of airborne particles; however, our inference of the particle's phase state at the time of collection is based on the shape the particle acquires at impaction on the substrate, which unlikely would change significantly within the ESEM chamber due to

adhesion forces between the particle and the substrate. These limitations need to be considered when interpreting our results."

**Moreover, as the reviewer suggested, we added the average temperature and RH at OMP during the sample periods to Table S1 and S2 and plotted the hourly variation of temperature and RH in Fig. S2 in revision:**

[Figure]

**Figure S2.** Hourly variation of temperature and relative humidity at OMP for available days. Shaded areas represent the sample collection periods.

2. Give the full name of "SEM" at Line 68 instead of at Line 71. Are SEM and ESEM the same?

**We are sorry that we did not make this clear. We used the same ESEM for both CCSEM-EDX and tilted imaging experiments. We revised line 158 and line 213 as below:**

RL167-168: "We utilized an environmental SEM (ESEM, Quanta 3D, Thermo Fisher) equipped with a FEI Quanta digital field emission gun, operated at 20 kV and 480 pA."

RL226-228: "We utilized tilted view imaging combined with the ESEM to estimate the phase state of particles based on their shapes. For each sample, we evaluated more than 150 randomly selected particles. Moreover, tilted view imaging and CCSEM-EDX experiments were performed independently."

3. Line 125. Change "Experimental" to Experiments.

**Change has been made.**

4. Line 177-178. "87 for S3 and 37 for S5 for Pico 2015, and 142 and 171 particles for S3-3 and S4-2 for Pico 2017". These data are not same as those in Table S2 and Table S3. Please check which are correct.

   **Thanks for this comment. We have revised the sentence as below:**

   > RL188-192: "Due to beamline time constraints for STXM analysis, we focused only on selected samples and a limited number of selected particles (653 for SA1, 208 for SA2, and 425 for SA3 for Pico 2014, 86 for S3 and 37 for S5 for Pico 2015, and 140 and 166 particles for S3-3 and S4-2 for Pico 2017)."

5. Line 190, I don't understand what TCA is proportional to?

   **We apologize for the confusion. TCA is proportional to the particle thickness based on Eqn. 1 and 2. We modified the sentence as follows:**

   > RL203-204: "Thus, TCA is proportional to the particle thickness, and it can be used as an indicator for particle thickness (O'Brien et al., 2014; Tomlin et al., 2020)."

6. Line 245. Explain how you determined the air mass source is wildfire from "CO source contributions".

   **PMO is usually dominated by outflows of anthropogenic emissions from North America and is occasionally affected by wildfire events. Wildfire events at PMO can contribute as much as anthropogenic emissions when they occur, enhancing CO by ~25 ppbv upon the North Atlantic background on average. In terms of aerosol composition, wildfire events have a clearer signature, such as enhanced BC, so we consider any samples with over 20% of FLEXPART CO from biomass burning would have a clear impact on the aerosol composition and properties of collected samples.**

7. Line 277-281, the data described for SA1, SA2 and SA3 are different from the corresponding data in Table 1.

   **We have revised the sentence as below:**

   > RL290-295: "Based on the CO tracer analysis, the major CO sources for SA1 were anthropogenic emissions in North America (~49 %), anthropogenic emissions in South America (~8 %), and wildfires in North America (~19 %). For SA2, the major CO sources were North American anthropogenic emissions (~42 %), African anthropogenic emissions (~16 %), and North American wildfires (~31 %). For SA3, anthropogenic (~49 %) and wildfire (~49 %) emissions in North America were the two major CO contributors (Lata et al., 2021)."

8. Line 284-286, "and S1, S3, and S6 were influenced by both anthropogenic and wildfire CO emissions in North America (~56 %, ~79 %, ~40 %, and ~59 % for anthropogenic CO source,

and ~42 %, ~19 %, ~53 %, and ~25 % for wildfire CO sources, respectively)." Check the values (there are four values for three samples).

**We have revised the sentence as below:**

RL296-299: "Based on the CO tracer simulations (Fig. S8), the major source of CO for sample S2 was anthropogenic emissions in North America (~84 %), and S1, S3, S5, and S6 were influenced by both anthropogenic and wildfires CO emissions in North America (~56 %, ~79 %, ~38 %, and ~59 % for anthropogenic CO sources, and ~42 %, ~19 %, ~53 %, and ~25 % for wildfires CO sources, respectively)."

9. Line 289. Change "Chemical-resolved" to "Chemically-resolved".

**Change has been made.**

10. Line 292. Change ">400 particles cm-3" to ">400 particles cm-3 "

**We made the change as suggested by the reviewer.**

11. Line 296. "Our particles are internally mixed based on tilted transmission electron microscopy (TEM, the titled angle was 70°) (Fig. S8).", Line 328. "This observation is consistent with their STXM images and tilted TEM images (Fig. S8)". Give a more detailed explanation how an internal mixing state is determined? Line 297. Change "titled angle" to "tilted angle".

**Thanks for bringing up this point. We can see internally mixed EC and inorganic inclusions coated by organics in representative tilted TEM images (Fig. S9 in revision) and STXM images of individual particles (Fig. 4). To make this clearer, we have revised the manuscript as below:**

RL316-317: "Tilted transmission electron microscopy (tilted angle 70°) images show that inorganic inclusions (e.g., sea salt, nitrate, sulfate, dust) are internally mixed and coated by organics (Fig. S9)."

**And:**

RL349-352: "Moreover, STXM images (see Fig. 4) indicate that particles are internally mixed and coated by organic species, suggesting our samples might be highly aged during transport in the FT (China et al., 2015; Motos et al., 2020). This observation is consistent with our tilted TEM images showing that EC and IN inclusions were internally mixed with organics (Fig. S9)."

12. Line 298. "Fig. 2(b to i) show" should be Fig. 3(b to i). g

**We have corrected it:**

RL317-319: "Figure 3a shows the average number fraction of different particle types in each sample, and Fig. 3b to 3i show chemically-specific normalized particle size distributions."

13. Line 304. "area equivalence diameter" . Do you mean "area equivalent diameter"?

> **We have revised the sentence:**
>
> > RL322-325: "Sea salt with sulfate particles with area equivalent diameters greater than 0.6 μm have been shown to be a product of aqueous phase processing (i.e., fog and cloud processing) (Ervens et al., 2011; Kim et al., 2019; Lee et al., 2011, 2012; Zhou et al., 2019), and those with area equivalent diameter less than 0.6 μm might have been generated from marine sources (Sorooshian et al., 2007; Yu et al., 2005)."

14. Line 306. The values of 79.6% and 1.1% did not match the values in Table S2.

> **We have corrected the numbers.**
>
> > RL325-327: "Sea salt and sea salt with sulfate particles dominated (~28.2 % and ~31.5 %, respectively) sample S3-1, with a smaller fraction of organic particles (OC and CNO, ~6.3 % and ~23.4 %, respectively) than in other samples."

15. Line 323. "Figure 4 shows the STXM-NEXAFS Carbon K-edge chemical speciation maps and spectra of four typical particle mixing states of OC (green), IN (blue), and EC (red) found in S3-3 and S4-2, which are (a) organic particle (green), (b) EC core (red) and coated by OC (green), (c) internally mixed EC (red) and In (cyan) coated by OC (green), and (d) In (cyan) coated by OC (green)". Do "blue" and "cyan" both indicate the inorganics? DO "IN" and "In" both indicate the inorganics? And the description here is different from the caption of Figure 4.

> **Sorry for the confusion. We used IN for inorganic components retrieved from STXM-NEXAFS spectroscopy and made changes accordingly in the revised manuscript. We also have corrected our manuscript and used cyan for IN in Fig. 4.**
>
> > RL344-347: "Figure 4 shows STXM-NEXAFS Carbon K-edge chemical speciation maps and spectra for four typical particle mixing states of OC (green), IN (cyan), and EC (red) found in S3-3 and S4-2, which are (a) organic particle (green), (b) EC core (red) and coated by OC (green), (c) internally mixed EC (red) and IN (cyan) coated by OC (green), and (d) IN (cyan) coated by OC (green)."
>
> We also corrected the caption of Fig. 4:
>
> > **"Figure 4. Representative STXM-NEXAFS spectra of (a) organic particle (green), (b) EC core (red) coated by OC (green), (c) internally mixed elemental carbon (red), and IN (cyan) core coated by OC (green), and (d) IN (cyan) core coated by OC (green) from Pico 2017 S3-3 and S4-2 samples. White scale bars represent 500 nm."**

16. Line 331. "all samples" Check Figure S9 is for the results of all samples of only seven samples.

**To make this sentence clearer, we revised it as below:**

> RL354-355: "The particle chemically-resolved size distributions for seven samples (SA1-SA3, S3, S5, S3-3, and S4-2) analyzed with STXM-NEXAFS are shown in Fig. S10."

17. Line 332. "S3-3 and S4-4 samples" Do you actually mean S3-3 and S4-2 samples? I do not see S4-4 in Figure S9, and in Table S2, the sample analyzed by STXM-NEXAFS is sample S4-1. Also check the values that did not match the ones in Table S2.

**Thanks for pointing this out. We have corrected the sentence and Table S2.**

> RL355-356: "In the S3-3 and S4-2 samples, OCIN particles are dominant (~87.8 % and ~98.8 %, respectively), and there is only a very small fraction of OC-rich particles (~5.2 % and ~1.2 %, respectively)."

18. Line 344. "Figure 5 shows violin plots of the 'corrected' aspect ratio (left) and representative tilted images (right) for Pico 2014 (a to c), Pico 2015 (d to i), and Pico 2017 (j to q)." The description here is different from the caption of Fig. 5. Correct it.

**We have corrected the caption of Fig. 5 as below:**

> **"Figure 5. Violin plots of 'corrected' aspect ratio (left) and typical tilted images (right) of Pico 2014 (a to c), Pico 2015 (d to i), and Pico 2017 (j to q). Distributions in the left panels of (d to i) are the aspect ratio of Pico 2015 particles collected on stages 3 (left) and 4 (right). The shaded region corresponds to the different phase states (red: solid state; green: semisolid state; blue: liquid state). The red lines indicate the means, and the black dots the medians."**

19. Line 368. "The substantial fraction of solid and semisolid particles might be less oxidized" In Table 1, I found that SA1 and S6, whose average aging time is both longer than 16 days, have smaller fraction of liquid particles than other samples. Can you explain why the fraction of liquid particles is smaller with longer aging time?

**We appreciate this great question from the reviewer. We wanted to provide a potential hypothesis for the phenomenon we observed, and future studies might need to confirm our hypothesis. We hypothesized that the substantial fraction of solid and semisolid particles might be less oxidized in the FT than the liquid particles during transport. We revised the statement as follows.**

> RL404-404: "Thus, we hypothesize that a substantial fraction of solid and semisolid particles might be less oxidized and less prone to be removed via aqueous-phase processes than liquid particles in the FT during transport."

20. Line 379. Change "5(a, d, e, I, and j to o))" to "5(a, d, e, i, and j to o))"

**The correction has been made as below:**

RL423-426: "In addition, we observed two different types of aspect ratio distributions: (a) narrow distribution with mean aspect ratios below 4 and a smaller fraction of particles (< 35 %) with aspect ratios greater than 4 (standard deviations of aspect ratio were ranging from 0.9 to 2.1) (Fig. 5a, 5d, 5e, and 5i to o), and (b) broad distribution with a larger fraction of particles (> 40 %) with aspect ratios greater than 4 (standard deviation of aspect ratio were ranging from 1.4 to 2.4) (Fig. 5b, 5c, 5f, 5g, 5h, 5p, and 5q)."

21. Line 383. "For S4-2, a possible reason is that the volatile and less viscous species of particles collected on the TEM grid have already evaporated and left these tiny residuals around those big particles (see Fig. 5(f) right panel) due to difference in temperature, RH, and pressure between OMP and SEM chamber." Does this problem also exist in the experiments of other samples?

**This is a great question. Based on the tilted images of all samples, we did not see other samples having the same issues since we did not observe tiny residuals around the particles. We added the discussion of limitations of the ESEM experiments in the revised manuscript RL168-174:**

RL168-174: "Ambient particle samples were analyzed with ESEM at 293 K, under vacuum conditions (~2×10-6 Torr) and therefore at RH values near zero, which might lead to losses of volatile and semivolatile materials. Moreover, the temperature and RH inside the ESEM chamber differed from those at the OMP during sample collections (Fig. S2). RH and T affect the phase state of airborne particles; however, our inference of the particle's phase state at the time of collection is based on the shape the particle acquires at impaction on the substrate, which unlikely would change significantly within the ESEM chamber due to adhesion forces between the particle and the substrate. These limitations need to be considered when interpreting our results."

**Moreover, we also revised the sentence below to make this point clearer to readers:**

RL430-433: "For S4-2, a possible reason is that the volatile and less viscous species in particles collected on the TEM grid have already evaporated and left these tiny residuals around those big particles (see Fig. 5f right panel) due to difference in temperature, RH, and pressure between OMP and the SEM chamber, a phenomenon which has not been observed in other samples. Hence, those particles remaining on the substrate in S4-2 have a higher viscosity than the original particles."

22. Line 402. I did not see the viscosity of BBOA predicted in DeRieux et al. (2018) is up to 10 [12] Pa s. I suggest you only show what is the range of the viscosity under the atmospherically relevant RH. Add Li et al. (2020) who also calculated the viscosity of BBOA based on volalilaty distributions (Li, Day et al. 2020).

**We want to thank the reviewer for this comment. The RH values at OMP during our study range from ~6% to ~67% (see Fig. S2 and Table S1 and S2 in revision), we think it should still be valid to report these literature values since they are within the RH range at OMP during our study. Moreover, we added a sentence to discuss the viscosity of BBOA reported by Li et al., 2020:**

RL449-454: "DeRieux et al., 2018 predicted the viscosity of biomass burning particles using their chemical composition, and they found that the viscosity of biomass burning particles varied between $10^{-2}$ Pa·s and $10^9$ Pa·s depending on the RH. Liu et al., 2021 found that ambient and lab-generated biomass burning particles are in a non-solid state at RH between 20 % and 50 %. Li et al., 2020 predicted the viscosity of ambient biomass burning organic aerosols from volatility distribution and found that it varies between $10^{-2}$ Pa·s (liquid state) to above $10^{12}$ Pa·s (solid state) in Athens (Greece) and $10^{-2}$ Pa·s (liquid state) to ~ $10^9$ Pa·s in Mexico City (Mexico)."

23. Line 416. "Shaded areas represent regions of different phase states (liquid: blue, semisolid: green, and solid: red), with the boundaries of each region based on (O'Brien et al., 2014)." Can you give a more detaied explaination how to get the boundary lines?

   **We appreciate the reviewer for this question. These boundary lines were based on previously reported field and lab-generated organic particles measurements described in O'Brien et al., 2014. To make this clear, we revised the sentence as below:**

   RL465-467: "Symbols are colored by their OVF. Shaded areas represent regions of different phase states (liquid: blue, semisolid: green, and solid: red), with the boundaries of each region based on measurements of field and lab-generated organic particles reported in O'Brien et al., 2014."

24. Line 430. "We used the density ($\rho_{org}$), hygroscopicity ($\kappa_{org}$), and dry glass transition temperature ($T_{g,org,0}$) of organic particles as reported by Schum et al., 2018 (see SI) since we do not have molecular compositions for our samples and Schum et al., 2018's samples were also collected at OMP during the same seasonal period (June and July).". The previous analysis in this manuscript mentioned that the composition of organic matter is quite different for different samples. Therefore, $T_{g,org,0}$ would be changed. There are three samples in the study of Schum et al. (2018), and the estimated Tg are also varied. Discussion of the uncertainties in Tg,org,0 is betted added.

   **We thank you for this valid comment. Please see our response to major comment 1. We added the discussion of uncertainties in the revised manuscript.**

25. Line 441, also cite (Schmedding, Rasool et al. 2020, Li, Carlton et al. 2021).

   **We have added these two references. Thanks for suggesting them.**

26. Line 490, cite (Li, Carlton et al. 2021, Shrivastava, Rasool et al. 2022).

   **We have added these two references. Thanks for suggesting them.**

27. Line 930. Change "solid black cycles" to "solid black circles"?

   **We have revised Fig. 2 and its caption as below:**

[Figure]

**Figure 2. (a) Normalized particle size distribution from 10 to 800 nm measured from SMPS measurements, and (b) SMPS derived total particle concentrations (left y-axis, red line) and CO tracer retrieved from FLEXPART simulations (right y-axis, blue line) from 05 to 21 July 2017.**

28. There is no need to use italics in the columns 12 and 13 in the first row in Table 1.

   **Thanks for this. We have corrected this issue.**

29. What does the colorbar in Figure 1 indicate?

   **We added the flowing sentience to the caption of Fig. 1 and Fig. S7 in revision:**

   **"The color bars indicate the ratio of column integrated residence time to the maximal residence time in the logarithmic scale. The X-axis and y-axis represent latitude and longitude, respectively."**

30. The inserted figures should be described in the caption of Figure 3.

   **Thanks for this comment. We revised the caption of Fig. 3 as below:**

   **"Figure 3. Chemically-resolved size distributions were inferred from the CCSEM-EDX data for 2017. (a) Fraction of different particle types for all samples. Normalized chemically-resolved size distributions of (b) S3-1, (c) S3-2, (d) S3-3, (e) S3-4, (f) S4-1, (g) S4-2, (h) S4-3, and (i) S4-4. Inserts represent the normalized number fraction of different particle types as a function of particle size."**

31. Change "SA1" to "SA2" for panel b in Figure 5.

   **Thanks for pointing this out. We have corrected this in Fig. 5.**

**Supporting Information :**

*32.* Line 2. *The title in the supplementary is different from the title in the manuscript.*

**We attempted to correct all the errors in the updated SI. We also corrected the title.**

33. Line 21. "where Tg,w is equal to 136 K, is the Tg for pure water". Cite (Kohl, Bachmann et al. 2005).

**We have corrected these two sentences as below:**

RSL51-52: "where $T_{g,w}$ is the $T_g$ for pure water, $k_{GT}$ is the Gordon-Taylor constant, $\kappa_{org}$ is the CCN-derived hygroscopicity parameter of the organic fraction, $\rho_{org}$ and $\rho_w$ are the density of water and organic material, respectively."

RSL58-60: "Moreover, $k_{GT}$, $T_{g,w}$, $\kappa_{org}$, and $\rho_{org}$ were assumed to be 2.5 (Shiraiwa et al., 2017), 136 K (Kohl et al., 2005), 0.12 (Schum et al., 2018), and 1.4 g cm$^{-3}$ (Schum et al., 2018), respectively."

34. Line 29. "Moreover, k$_{GT}$, T$_{g,w}$, $\kappa_{org}$, and $\rho_{org}$ were assumed to be 2.5 (Shiraiwa et al. 2017), 309 K (Schum et al. 2018), 0.12 (Schum et al. 2018), and 1.4 g cm-3 (Schum et al. 2018), respectively." Why 309 K is for Tg,w? Check it.

**We have corrected this as below:**

RSL58-60: "Moreover, $k_{GT}$, $T_{g,w}$, $\kappa_{org}$, and $\rho_{org}$ were assumed to be 2.5 (Shiraiwa et al., 2017), 136 K (Kohl et al., 2005), 0.12 (Schum et al., 2018), and 1.4 g cm$^{-3}$ (Schum et al., 2018), respectively."

35. Figure S2. What the x-axis stands for in figures b to r?

**The X-axis shows the particle number. To make this clear, we revised the caption of Fig. S2 (Fig. S3 in revision) as below:**

**"Figure S3. Relative element percentage of 15 elements (C, N, O, Na, Mg, Al, Si, P, S, Cl, K, Ca, Mn, Fe, Zn) for (a) average relative atomic ratios for all samples, (b) SA1, (c) SA2, (d) SA3, (e) S1, (f) S2, (g) S3, (h) S4, (i) S5, (j) S6, (k) S3-1, (l) S3-2, (m) S3-3, (n) S3-4, (o) S4-1, (p) S4-2, (q) S4-3, (r) S4-4. The X axis indicates the particle number."**

36. In Figure S2 and Figure S3, are the relative atomic ratios of elements same as the relative element weight?

**We are sorry for the confusion. To make it clear, we changed that to element percentage and modified the entire manuscript to keep it consistent. Thus, the caption of Fig. S3 (Fig. S4 in revision) has been revised as:**

**"Figure S4. Flow chart to classify Pico 2017 particle types based on their element percentage retrieved from CCSEM/EDX measurements."**

37. Figure S4. Change "Jun" to "June".

   **Thanks for pointing this out. We have revised that.**

38. In Fig. S5-S6, I don't understand why the residence time is in percentage and how did you calculate it?

   **In these results, residence time has been integrated vertically for the entire transport time (20 days) and the whole atmosphere. The residence time shown is color-coded in the logarithmic scale representing its ratio to the location of maximal integrated residence time (100 %). The reason for doing this is to simplify the comparison between two transport cases because the value of maximal integrated residence time for each transport case can be largely different. We can easily tell the relative time a plume spends over land vs. ocean and have a clear view of the transport pathway. To make this clear, we revised the caption of Fig. S5 and S6 (Fig. S6 and S7 in revision) as below:**

   **"Figure S6. The vertical distribution of the retroplumes residence time at given upwind times retrieved from FLEXPART retroplumes for (a) S3-1, (b) S3-2, (c) S3-3, (d) S3-4, (e) S4-1, (f) S4-2, (g) S4-3, and (h) S4-4 for Pico 2017. The color bar represents the ratio of residence time to the highest residence time across the height scale at each upwind time. The black lines indicate the average height of the plumes during transport."**

   **"Figure S7. Column-integrated residence time over the 20-day transport time and the vertical distribution of the retroplumes residence time at given upwind times retrieved from FLEXPART retroplumes for Pico 2015. (a, b) S1, (c, d) S2, (e, f) S3, (g, h) S4, (i, j) S5, (k, l) S6. For panels a, c, e, g, and i, the color bars indicate the ratio of column integrated residence time to the maximal residence time at each upwind time in the logarithmic scale, and the X-axis and y-axis represent latitude and longitude, respectively. For panels b, d, f, h, j, and l, the color bars represent the ratio of residence time to the highest residence time across the height scale at each upwind time, and the black lines indicate the average height of the plumes during transport."**

39. In the caption of Figure S10, "Mean ambient temperature (blue) and the predicted RHdependent $T_{g,org}$ values (green)". The ambient T is actually in green and $T_{g,org}$ is in blue in the figure.

**Thanks for pointing this out. We have revised the caption of Fig. S11:**

"**Figure S11. Mean ambient temperature (green) and relative humidity (RH) (red) extracted from the GFS analysis along the FLEXPART modeled path weighted by the residence time and the predicted RH-dependent $T_{g,org}$ values (blue) for (a) S3-1, (b) S3-2, (c) S3-3, (d) S3-4, (e) S4-1, (f) S3-2, (g) S4-3, and (h) S4-4. The blue and red shaded areas represent one standard deviation of ambient temperature and RH from the GFS analysis. The green shaded areas represent uncertainties of predicted $T_{g,org}$ estimated from the range of $T_{g,org}(RH = 0\%)$ and uncertainties in RH.**"

**References**

Cheng, Y., H. Su, T. Koop, E. Mikhailov and U. Pöschl (2015). "Size dependence of phase transitions in aerosol nanoparticles." Nature Communications 6: 5923.

Kohl, I., L. Bachmann, A. Hallbrucker, E. Mayer and T. Loerting (2005). "Liquid-like relaxation in hyperquenched water at [less-than-or-equal]140 K." Physical Chemistry Chemical Physics 7(17): 3210-3220.

Li, Y., A. G. Carlton and M. Shiraiwa (2021). "Diurnal and Seasonal Variations in the Phase State of Secondary Organic Aerosol Material over the Contiguous US Simulated in CMAQ." ACS Earth and Space Chemistry.

Li, Y., D. A. Day, H. Stark, J. L. Jimenez and M. Shiraiwa (2020). "Predictions of the glass transition temperature and viscosity of organic aerosols from volatility distributions." Atmos. Chem. Phys. 20(13): 8103-8122.

Li, Y. and M. Shiraiwa (2019). "Timescales of secondary organic aerosols to reach equilibrium at various temperatures and relative humidities." Atmos. Chem. Phys. 19(9): 5959-5971.

Petters, M. and S. Kasparoglu (2020). "Predicting the influence of particle size on the glass transition temperature and viscosity of secondary organic material." Scientific Reports 10(1): 15170.

Petters, S. S., S. M. Kreidenweis, A. P. Grieshop, P. J. Ziemann and M. D. Petters (2019). "Temperature- and humidity-dependent phase states of secondary organic aerosols." Geophys. Res. Lett. 46.

Richards, D. S., K. L. Trobaugh, J. Hajek-Herrera, C. L. Price, C. S. Sheldon, J. F. Davies and R. D. Davis (2020). "Ion-molecule interactions enable unexpected phase transitions in organic-inorganic aerosol." Science Advances 6(47): eabb5643.

Schmedding, R., Q. Z. Rasool, Y. Zhang, H. O. T. Pye, H. Zhang, Y. Chen, J. D. Surratt, F. D. Lopez-Hilfiker, J. A. Thornton, A. H. Goldstein and W. Vizuete (2020). "Predicting secondary organic aerosol phase state and viscosity and its effect on multiphase chemistry in a regional-scale air quality model." Atmos. Chem. Phys. 20(13): 8201-8225.

Shrivastava, M., Q. Z. Rasool, B. Zhao, M. Octaviani, R. A. Zaveri, A. Zelenyuk, B. Gaudet, Y. Liu, J. E. Shilling, J. Schneider, C. Schulz, M. Zöger, S. T. Martin, J. Ye, A. Guenther, R. F. Souza, M. Wendisch and U. Pöschl (2022). "Tight Coupling of Surface and In-Plant Biochemistry and Convection Governs Key Fine Particulate Components over the Amazon Rainforest." ACS Earth and Space Chemistry 6(2): 380-390.

---

## Referee Report (RR1)

In general, the author's answer and revised sentences are much clearer than before. However, I still need some clarification regarding the authors' responses to resolve some of the possible confusion. I have several more comments that the authors should address before publication of this manuscript. Please see below.

1. I confused on the definition of the phase state of the field samples. It must be contain various phases inside the particles as shown in the figures. Do you mean the particle is overall solid, semisolid or liquid? A clear definition of the concepts for the real aerosol particles in the introduction should be added.

2. The authors mentioned that the samples were analyzed less than one year after collection. One year maybe enough time to change the chemical compositions and related phase states of the collected samples. Can you provide some previous work to solve this issue?

3. Still I am confusing how you described/concluded the phases for the filter samples collected from FT even the experimental condition was different. During the experiment, the experimental conditions of RH and temperature were significantly different compared to the environment conditions of FT (higher RH and lower temperature). This should be stated clearer.

---

## Author Response (AR2)

We want to thank the reviewers for their further comments. Below, we list each reviewer's comment (regular font), followed by our response (indented, **bold** font), followed by corresponding changes in the revised manuscript (indented, blue font). RL represents the line number in the revised version.

**Anonymous Referee #1**

This manuscript by Cheng et al. has been improved and been revised accordingly based on the reviewers' comments. I support the publication of this manuscript on this journal. I have two technical comments.

1.      Abbreviations are defined several times in the manuscript (e.g., SEM, STXM-NEXAFS, and FRXPART). Please check them.

2.      Figure S7. The last panel should be "l" instead of "i"

> **We appreciate the reviewer for providing us with these comments. We have revised the manuscript and SI based on these comments.**

**Anonymous Referee #2**

In general, the author's answer and revised sentences are much clearer than before. However, I still need some clarification regarding the authors' responses to resolve some of the possible confusion. I have several more comments that the authors should address before publication of this manuscript. Please see below.

> **We appreciate the constructive feedback from the reviewer. We attempted to clarify further and noted the study's limitations. Below are our responses to each comment:**

**Comments:**

1.  I confused on the definition of the phase state of the field samples. It must be contain various phases inside the particles as shown in the figures. Do you mean the particle is overall solid, semisolid or liquid? A clear definition of the concepts for the real aerosol particles in the introduction should be added.

> > **We thank the reviewer for pointing this out and agree that it should be clearly defined in the manuscript. As the reviewer noted, we study the overall phase state of individual (internally mixed) FT particles. Based on our knowledge, only a limited number of studies investigated ambient particle phase state (e.g., Bateman et al., 2016; Liu et al., 2017, 2021; Pajunoja et al., 2016; Slade et al., 2019), and all of them investigated the phase state of entire particles. We agree that the phase state inside the particles (e.g., inorganic inclusions) can be different, causing phase separation. We added the following sentence in our introduction to specify that we are looking at the overall phase state of particles:**
> >
> > > RL117-119: "In this study, we present an overview of the overall phase state of individual FT atmospheric aerosol particles (internally mixed) collected at OMP over three different years, which are July 2014 (Pico 2014), June and July 2015 (Pico 2015), and 2017 (Pico 2017)."

2.  The authors mentioned that the samples were analyzed less than one year after collection. One year maybe enough time to change the chemical compositions and related phase states of the collected samples. Can you provide some previous work to solve this issue?

> **In the previous response, we noted the potential modification of particles due to storage (comment 4 in the previous response). It would be fantastic if we could analyze our samples right after collecting them. However, that is often not feasible to accomplish for offline measurements of samples collected at the remote site since these sites are usually difficult to access, sample delivery time can be long, and there are always limited instruments and labor time to analyze them, which makes these type of research typically publish their results few years after the sample collection (see (Allen et al., 2021; China et al., 2017; Cozic et al., 2008; Dzepina et al., 2015; Lata et al., 2021; Moffet et al., 2010; Schum et al., 2018; Zhang et al., 2013). As the reviewer noted, we cannot exclude with certainty that some transformations might have taken place between the sampling and the analysis times. This is a limitation of any offline analysis of field-collected samples. We have tried to minimize oxidation and photolysis by wrapping the sample with Al foil and kept in zip lock bags immediately after collection, which is a typical method for storing field-collected aerosol samples** (e.g., Adachi and Buseck, 2011; Kirillova et al., 2016; Marsh et al., 2017; Stockwell et al., 2016). **Although we cannot avoid potential particle modification due to storage, we believe our results still provide essential scientific findings for aerosol research in the phase state of the FT particles. Future studies should focus on sample storage strategy for offline measurements. To make these clearer, we revised the previous revised sentence as below:**

>> RL140-147: "Samples were placed in dedicated storage boxes wrapped in Al foil and kept in zip lock bags immediately after collection to avoid exposure to light and outside air, which is a typical sample storage strategy for field-collected aerosol samples (e.g., Adachi and Buseck, 2011; Kirillova et al., 2016; Marsh et al., 2017; Stockwell et al., 2016). The samples were then stored at ambient conditions to reduce the chances of particle modifications and oxidation that might have partially intercurred. However, we cannot exclude with certainty that some of such transformations might have taken place between the sampling and the analysis times. This is a limitation of any offline analysis of field-collected samples. We underline that the site is quite difficult to access; therefore, samples were delivered and analyzed as soon as it was feasible (less than one year after collection)."

3.  Still I am confusing how you described/concluded the phases for the filter samples collected from FT even the experimental condition was different. During the experiment, the experimental conditions of RH and temperature were significantly different compared to the environment conditions of FT (higher RH and lower temperature). This should be stated clearer.

> **Thanks for this comment. As we explained in our response to the reviewer's comment 1 in the first reviewer comments document, we agree there might be some potential modification of the particle phase inside the ESEM chamber due to vacuum and dry conditions, which is a common caveat of this method. Based on that, we also revised the manuscript in our revision. We agree that the temperature and RH were different at the site compared to those during our offline experiments, in which the temperature is usually about 10 K higher, and RH is usually about 6-**

**67% lower. The higher temperature can reduce viscosity during experiments, while lower RH can increase viscosity as a competition effect. However, we mentioned this caveat in the previous response, *"our inference of the particle's phase state at the time of collection is based on the shape the particle acquires at impaction on the substrate, which unlikely would change significantly within the ESEM chamber due to adhesion forces between the particle and the substrate."* We confirmed this by collecting sucrose and ammonium sulfate mixture (50/50 wt%) at 83% RH. Based to** (Tong et al., 2022)**, sucrose and ammonium sulfate mixture should be liquid at 83% RH and semisolid at 0% RH. As shown in Fig. R1, these sucrose and ammonium sulfate mixture particles have an average aspect ratio of about 3.15±0.27, indicating they are still in the liquid state under vacuumed conditions, which validates that the shape of organic particles after impact on the substrate would unlikely change significantly within the ESEM chamber. However, to make the potential change in particle phase state clearer, we revised the following sentence:**

> RL169-175: "Ambient particle samples were analyzed with ESEM at 293 K, under vacuum conditions (~$2\times10^{-6}$ Torr) and therefore at RH values near zero, which might lead to losses of volatile and semivolatile materials. Moreover, the temperature and RH inside the ESEM chamber differed from those at the OMP during sample collections (about 10 K higher and 6-67% lower, respectively, see Fig. S2). RH and T affect the phase state of airborne particles; however, our inference of the particle's phase state at the time of collection is based on the shape the particle acquires at impaction on the substrate, which unlikely would change significantly within the ESEM chamber due to adhesion forces between the particle and the substrate. These limitations need to be considered when interpreting our results."

[Figure]

**Figure R1. ESEM tiled image of 50/50 wt% sucrose and ammonium sulfate mixture particles collected on Carbon Type-B TEM grids at 83%RH.**